# Aberrant (pro)renin receptor expression induces genomic instability in pancreatic ductal adenocarcinoma through upregulation of SMARCA5/SNF2H

Yuki Shibayama et al.[#]

(Pro)renin receptor [(P)RR] has a role in various diseases, such as cardiovascular and renal disorders and cancer. Aberrant (P)RR expression is prevalent in pancreatic ductal adenocarcinoma (PDAC) which is the most common pancreatic cancer. Here we show whether aberrant expression of (P)RR directly leads to genomic instability in human pancreatic ductal epithelial (HPDE) cells. (P)RR-expressing HPDE cells show obvious cellular atypia. Whole genome sequencing reveals that aberrant (P)RR expression induces large numbers of point mutations and structural variations at the genome level. A (P)RR-expressing cell population exhibits tumour-forming ability, showing both atypical nuclei characterised by distinctive nuclear bodies and chromosomal abnormalities. (P)RR overexpression upregulates SWItch/ Sucrose Non-Fermentable (SWI/SNF)-related, matrix-associated, actin-dependent regulator of chromatin, subfamily a, member 5 (SMARCA5) through a direct molecular interaction, which results in the failure of several genomic stability pathways. These data reveal that aberrant (P)RR expression contributes to the early carcinogenesis of PDAC.

[#]A list of authors and their affiliations appears at the end of the paper.

Pancreatic ductal adenocarcinoma (PDAC) has a high rate of malignancy, with median survival of 6 months and 5-year survival remaining <10%[1]. Somatically acquired mutations become a driving force to promote the progression of PDAC. On average, a PDAC patient acquires 67 nonsynonymous mutations[2]. In particular, activating mutations of *KRAS* are almost ubiquitous and inactivating mutations of *TP53*, *SMAD4*, *CDKN2A*, genes related to chromatin modification are also prevalent among PDAC patients[2–4]. Recent whole-genome analyses have detected large numbers of structural variations in PDAC[4,5]. However, molecular mechanisms responsible for genome instability remains unclear.

Recently, we and others have shown that (pro)renin receptor [(P)RR] expression is aberrant in human PDAC tissues, irrespective of the PDAC stage[6,7]. Furthermore, higher (P)RR expression was associated with severe stage in the TNM Classification of Malignant Tumours (TNM)[6]. Interestingly, aberrant (P)RR expression was synchronized with the appearance of atypical nuclei observed in pancreatic intraepithelial neoplasia (PanIN)-2 and PanIN-3 lesions[6,7], suggesting a potential role of (P)RR in genomic homeostasis.

(P)RR plays a role of multiple cellular functions, such as renin-angiotensin system, Wnt signalling pathway, extracellular and vesicular acidification, cell polarity and maintenance of the pyruvate dehydrogenase (PDH) complex[8]. The specific binding of prorenin and renin to the extracellular domain of (P)RR cleaves angiotensin I from angiotensinogen thus activating renin-angiotensin system[9]. It has also been shown that intracellular signals such as mitogen-activated protein (MAP) kinase are activated by ligands, independent of renin-angiotensin system[9–11]. Moreover, the extracellular domain of (P)RR also has a molecular interaction with low-density lipoprotein receptor protein 6 (LRP6) and Frizzled 8 of Wnt receptor complex[12]. The activation of Wnt signalling pathway through this molecular interaction is related to the development of PDAC[7,13], Glioma[14]. and colorectal cancer[15]. (P)RR plays a role in an accessory protein of vacuolar $H^+$-ATPase (V-ATPase)[16], which mediates the acidification of the extracellular space or vesicles to facilitate membrane trafficking and fusion, receptor-mediated endocytosis and autophagic protein degradation[17]. (P)RR is also involved in partitioning defective homologue 3 (Par3) system responsible for apical-basal orientation[18] and noncanonical Wnt planar cell polarity (PCP) pathway for proximal-distal orientation[19,20] in the cellular polarity. Binding of the (P)RR to PDH E1β-subunit (PDHB) contributes to the maintenance of aerobic glucose metabolism[21]. However, whether these cellular functions dominantly affect the genomic instability under aberrant (P)RR expression remains to be solved.

In the present study, we explore whether aberrant (P)RR expression directly induces genomic instability responsible for the generation of tumour heterogeneity in human pancreatic ductal epithelial (HPDE) cells, which are capable of transforming into PDAC[22]. To determine the level of genomic instability associated with aberrant (P)RR expression, we performed whole-genome sequencing analysis for (P)RR-expressing HPDE cells. Using nanoscale liquid chromatography with tandem mass spectrometry (LC-MS/MS), we also elucidated the molecular mechanism of genomic instability induced by aberrant (P)RR expression. Through these analyses, we demonstrate that aberrant (P)RR expression directly induces genomic instability by the disorder of SWItch/Sucrose Non-Fermentable (SWI/SNF)-related, matrix-associated, actin-dependent regulator of chromatin, subfamily a, member 5 (SMARCA5) as a chromatin remodeller at the whole-genome level. The data reveal that inappropriate augmentation of (P)RR expression contributes to the genetic evolution of PDAC.

## Results

**Aberrant (P)RR expression generates diverse cellular atypia.** We established (P)RR-overexpressing HPDE-1/E6E7[23] and -HPDE-6/E6E7[24] cells using a stable vector inserted with the coding sequence of *ATP6ap2* encoding (P)RR to perform the following experiments. As a control, we included cells transfected by a vector without the insertion of *ATP6ap2*, referred to as Mock (Fig. 1a). The insertion of *ATP6ap2* was confirmed in both cells at six-passage by PCR (Fig. 1a). Compared to cells with Mock, (P)RR overexpression significantly induced much larger and variable cell area ($N = 100$ for each; Student's *t*-test; $P < 0.0001$ vs. Mock) in HPDE-1/E6E7 and HPDE-6/E6E7 cell population (Fig. 1b). (P)RR overexpression also exhibited nuclei with distinctive nuclear bodies (Fig. 1b). To evaluate the level of atypical nuclei in HPDE-1/E6E7 and HPDE-6/E6E7 cells, we performed Papanicolaou stain. In both cell populations under aberrant (P)RR expression, multi-nucleated cells were considerable and the increase of (P)RR expression significantly led to larger and variable nuclear area ($N = 100$ for each; Student's *t*-test; $P < 0.0001$ vs. Mock; Fig. 1c and Supplementary Fig. 1). Chromosome abnormality was also observed in both cell populations expressing (P)RR (Fig. 1c and Supplementary Fig. 1). It is clear from the data that (P)RR-expressing HPDE-1/E6E7 and HPDE-6/E6E7 cell population generates cells harboring diverse atypical nuclei.

**Genomic instability in HPDE cells with (P)RR overexpression.** To compare genomic instability between HPDE-1/E6E7 cell population with transient and stable (P)RR expression, we also established (P)RR-overexpressing HPDE-1/E6E7 cells using a stable and non- replicative transient vector inserted with the coding sequence of *ATP6ap2*[25] (Fig. 2a). The insertion of *ATP6ap2* was confirmed in HPDE-1/E6E7 cells with transient (P)RR expression by PCR (Fig. 2a).

We performed whole-genome sequencing for untreated, transient Mock- and (P)RR-overexpressing and stable Mock- and (P)RR-overexpressing HPDE-1/E6E7 cells. By comparing each Mock- and (P)RR-overexpressing cells against untreated HPDE-1/E6E7 cells, we detected point mutations, short insertions and deletions (indels), and structural variations (SVs). Our analyses identified much larger numbers of point mutations and SVs in stable (P)RR-overexpressing cells than other treated cells. Furthermore, stable (P)RR expression against transient (P)RR considerably induced higher numbers of somatic mutations and SVs than stable Mock expression against transient Mock. These data indicate that the level of (P)RR expression affects the difference in genomic instability (Fig. 2b, c and Supplementary Fig. 2 and Supplementary Data 1). Stable (P)RR-overexpressing cells increased the number of point mutations and SVs than stable Mock cells by 6.5- and 8.8- fold, respectively. Chromosomal translocations detected in the stable (P)RR-overexpressing cells numbered 122 and dominated in all the SVs (48%). However, there was no difference in the number of short indels among treated cells.

We next focused on protein alternating mutations in the stable (P)RR-overexpressing cells. Our analyses identified 63 nonsynonymous mutations, and known driver genes defined by the COSMIC database (https://cancer.sanger.ac.uk/cosmic; *FGFR3*, *MLL3*, *BRIP1* and *MSH6*) were included (Supplementary Data 2). In the SVs, breakpoints were detected in 5 COSMIC driver genes (*PDE4DIP*, *THRAP3*, *FANCD2*, *MDS1* and *CBL*; Supplementary Data 3). In addition, a chromosomal translocation was detected in a region close to a LINE1 transposable element located on intron1 of *TTC28*. Recurrent chromosomal translocation events in this region were reported in various cancers and were considered to be caused by LINE1 transposition[26–28]. Another chromosomal translocation was observed in *MACROD2* gene,

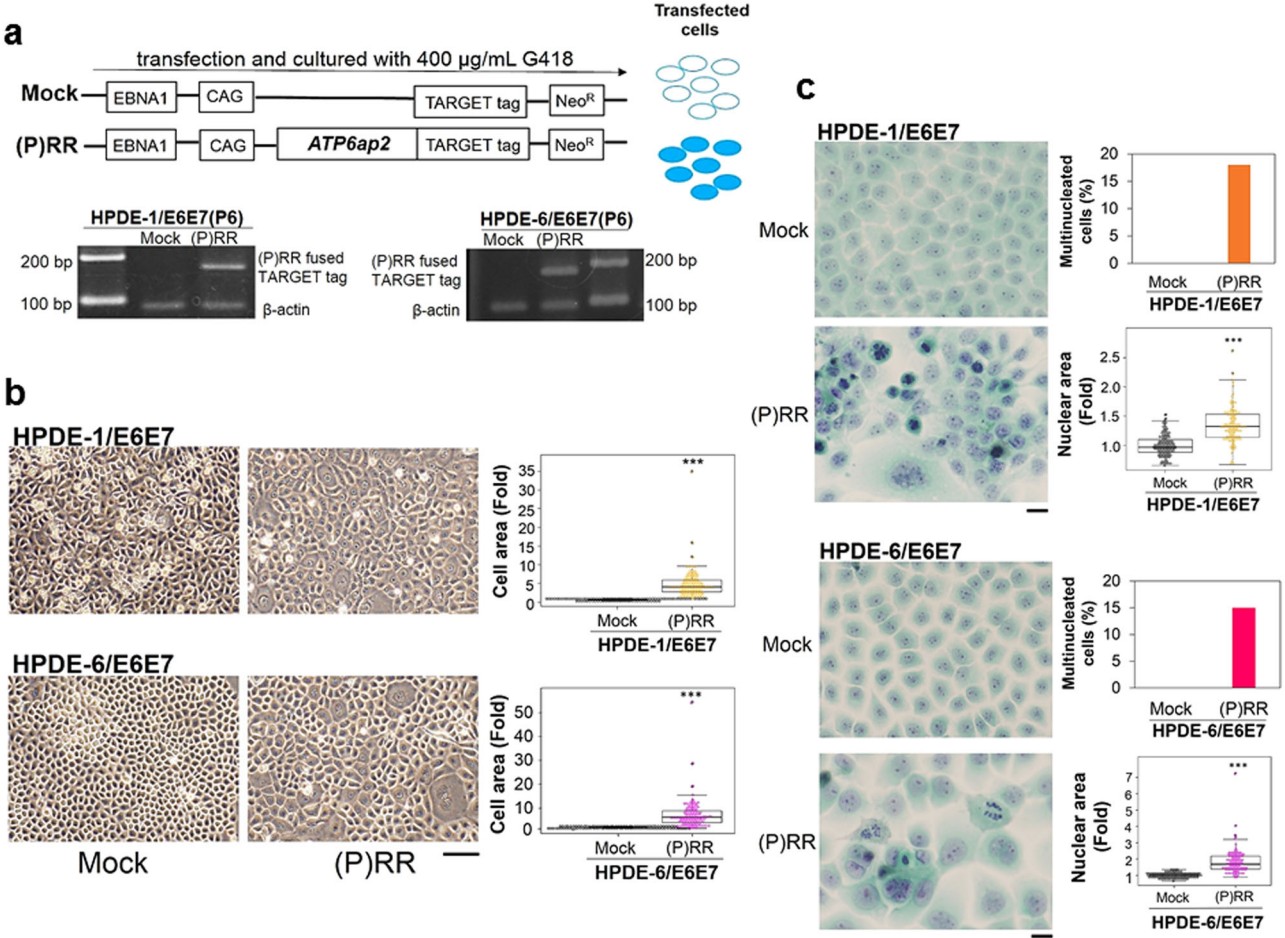

**Fig. 1 (P)RR-expressing HPDE-1/E6E7 and HPDE-6/E6E7 cell population exhibited cellular atypia. a** Upper: Stably maintained vector containing *ATP6ap2*, which encodes (pro)renin receptor [(P)RR]. The inclusion of the *EBNA1* gene enables this vector to be transferred to daughter cells at each round of cell division. Lower: Detection of (P)RR fused TARGET tag in human pancreatic ductal epithelial (HPDE)-1/E6E7 and HPDE-6 /E6E7 cells at six-passage. **b** Upper left: Representative image of HPDE-1/E6E7 cells expressing either Mock or (P)RR at six-passage under a phase-contrast microscope (×50). Upper right: The cell area in HPDE-1/E6E7 cells expressing either Mock or (P)RR. Averaged value of Mock cells is considered as 1 ($N = 100$ for each, ***$P < 0.0001$ vs. Mock). Lower left: Representative image of HPDE-6/E6E7 cells expressing either Mock or (P)RR at six-passage. Lower right: The cell area in HPDE-6/E6E7 cells expressing either Mock or (P)RR ($N = 100$ for each, ***$P < 0.0001$ vs. Mock). **c** Papanicolaou stain (×400). Upper left: Representative image of atypical nuclei in HPDE-1/E6E7 cells expressing either Mock or (P)RR at six-passage. Upper right: The percentage of multinucleated cells and the nuclear area in HPDE-1/E6E7 cells expressing either Mock or (P)RR. Averaged value of Mock cells is considered as 1 ($N = 100$ for each, ***$P < 0.0001$ vs. Mock). Lower left: Representative image of atypical nuclei in HPDE-6/E6E7 cells expressing either Mock or (P)RR at six-passage. Lower right: The percentage of multinucleated cells and the nuclear area in HPDE-6/E6E7 cells expressing either Mock or (P)RR. Averaged value of Mock cells is considered as 1 ($N = 100$ for each, ***$P < 0.0001$ vs. Mock). The horizontal line inside the box plot is the median and the vertical lines protruding the box extend to the minimum and the maximum values, respectively. The vertical width of the central box shows the inter-quartile deviation.

which is located at a fragile site and can be related with DNA repair system through PARP1 poly ADP-ribosylation (PARylation)[29,30]. These results suggest that (P)RR overexpression induces mutations in some important driver genes, which may contribute to genetic abbreviation in cancer.

**(P)RR cell population exhibits tumour-forming ability**. We examined whether implanted (P)RR-expressing HPDE-1/E6E7 cells possess pre-cancerous ability in vivo. For this purpose, 14-passage cells expressing either Mock or (P)RR were implanted into renal subcapsules of immunodeficient mice. The results from Sanger DNA sequencing and PCR products showed that mutations in *KRAS* codon 12[2–4] and in *CDKN2A* were absent in both Mock- and (P)RR-expressing cells (Fig. 3a). However, the (P)RR-expressing HPDE-1/E6E7 cell population began to form a tumour at 5 weeks after implantation, although explosive cell population growth was absent, owing to the absence of mutations in *KRAS*

codons. On the other hand, visible expansion was entirely absent upon the implantation of Mock cells (Fig. 3b, c). Histological analysis revealed that tissues formed by (P)RR-expressing cell implantation were composed of atypical cells with different shapes (Fig. 3d, e and Supplementary Fig. 3). Furthermore, these cells were also characterised by atypical and swollen nuclei with distinctive nuclear bodies and chromosomal abnormalities, such as bridge and fusion of the breakage-fusion-bridge (BFB) cycle induced by telomere dysfunction[31] (Fig. 3d, e). These results demonstrate that the (P)RR-expressing HPDE-1/E6E7 cell population exhibits tumour-forming ability in vivo.

**Intracellular localization of (P)RR**. To elucidate the intracellular localization of each domain of (P)RR after cleavage by a protease such as furin[7,32], we conducted immunofluorescence in the human PDAC cell lines PK-1, PANC-1 and MiaPaCa-2 and (P)RR-expressing HPDE-1/E6E7 cells using two anti-*ATP6AP2*

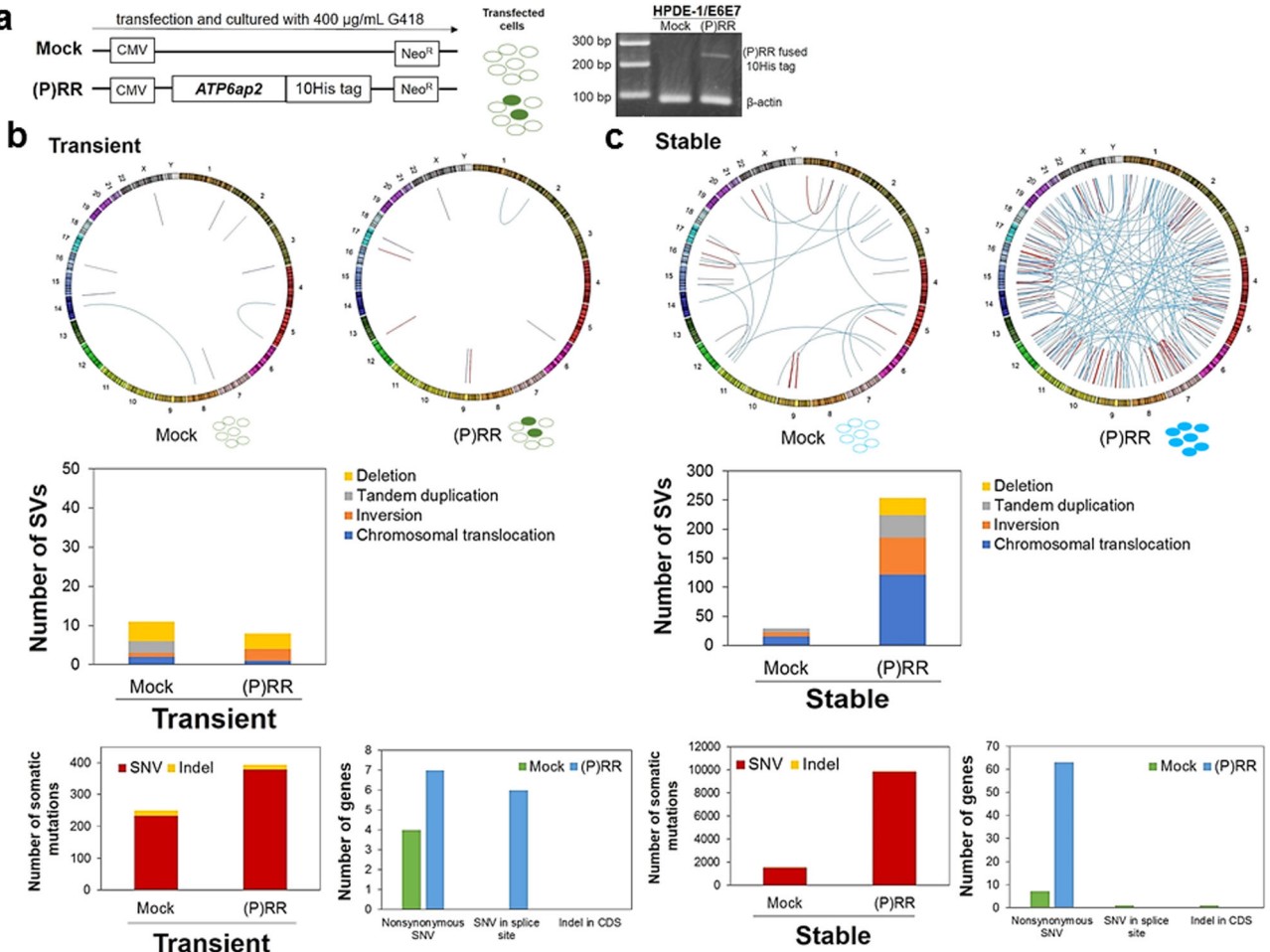

**Fig. 2 Genomic instability of HPDE-1/E6E7 cell population with transient and stable (P)RR expression. a** Left: Vector constructs for stable and non-replicative transient *ATP6ap2* encoding (pro)renin receptor [(P)RR] expression. Transfected cells were cultured with G418 for 21 days and analyzed after one passage. Right: Detection of (P)RR fused 10His tag in HPDE-1/E6E7 cells. **b** Upper: Circos plot showing distribution of SVs in transient Mock- and (P)RR-expressing cell population. Middle: Number of each SV in transient Mock-and (P)RR-expressing cell population. Lower: Total number of somatic mutations and mutated genes of the exome in transient Mock- and (P)RR-expressing cell population. **c** Upper: Circos plot showing distribution of SVs in stable Mock- and (P)RR-expressing cell population. Middle: Number of each SV in stable Mock- and (P)RR-expressing cell population. Lower: Total number of somatic mutations and mutated genes of the exome in stable Mock- and (P)RR-expressing cell population. SNV Single nucleotide variant; Indel Insertion/deletion; CDS Coding sequence.

antibodies that recognize different domains of (P)RR. Anti-*ATP6AP2* antibody interacts with an immunogen corresponding to a region within amino acids (a.a.) 146 and 350, covering from the extracellular to cytoplasmic domains of (P)RR, which detects all the full-length (P)RR [FL (P)RR], N-terminal fragment (NTF) and C-terminal fragment (CTF). The other antibody recognizes the region from a.a. 224 to 237, located in the extracellular domain of (P)RR[33], which detects both FL (P)RR and NTF. The anti-*ATP6AP2* antibody revealed that (P)RR is consistently expressed in both cytoplasm and nucleus in four different cell lines (Fig. 4a). In contrast, anti-*ATP6AP2* antibody recognizing the extracellular domain of (P)RR found that (P)RR expression is limited in the cytoplasm (Fig. 4b). These findings suggest that CTF of (P)RR is expressed in the nucleus. To further investigate the actual localization of CTF of (P)RR in the nucleus, we separated the cytoplasmic, soluble, and insoluble nuclear fractions from whole-cell lysates. In two different human PDAC cell lines, FL (P)RR is present in both the whole-cell lysates and the cytoplasmic fraction, whereas the CTF of (P)RR is dominant in the insoluble nuclear fraction occupied by nuclear lamina and chromatin-binding proteins using anti-*ATP6AP2* antibody recognizing from a.a. 146 to 350 (Fig. 4c).

**Downregulation of genomic stability pathways**. To elucidate the molecular mechanism responsible for the genomic instability induced by aberrant (P)RR expression, we performed LC-MS/MS analysis in the fraction of insoluble nucleus extracted from Mock- and (P)RR-expressing HPDE-1/E6E7 cells. From this analysis, we detected differences in expression of a total of 14,583 peptides. We implemented the global permutation-based false discovery rate (FDR) approach in Perseus[34] and 2340 peptides with FDR = 0.05 and minimal fold change (S0) = 0.5 were considered as significantly differentiated peptides (Supplementary Fig. 4a, b). Based on the fact that CTF of (P)RR was expressed in the insoluble nucleus containing chromatin-binding protein, further analyses were performed by using biological information regarding chromatin and/or DNA binding as Gene Ontology (GO) terms[35] (Supplementary Data 4). Using significant peptides filtered by GO terms, we also performed Ingenuity Pathway Analysis (IPA) to identify related diseases, and each molecular function and canonical pathways affected by the increase of (P)RR expression. Regarding downregulated molecules, cancer was identified as top of related diseases and it was not expected from the information of upregulated molecules (Fig. 5a and Supplementary Fig. 4c). By analyses of upregulated and downregulated

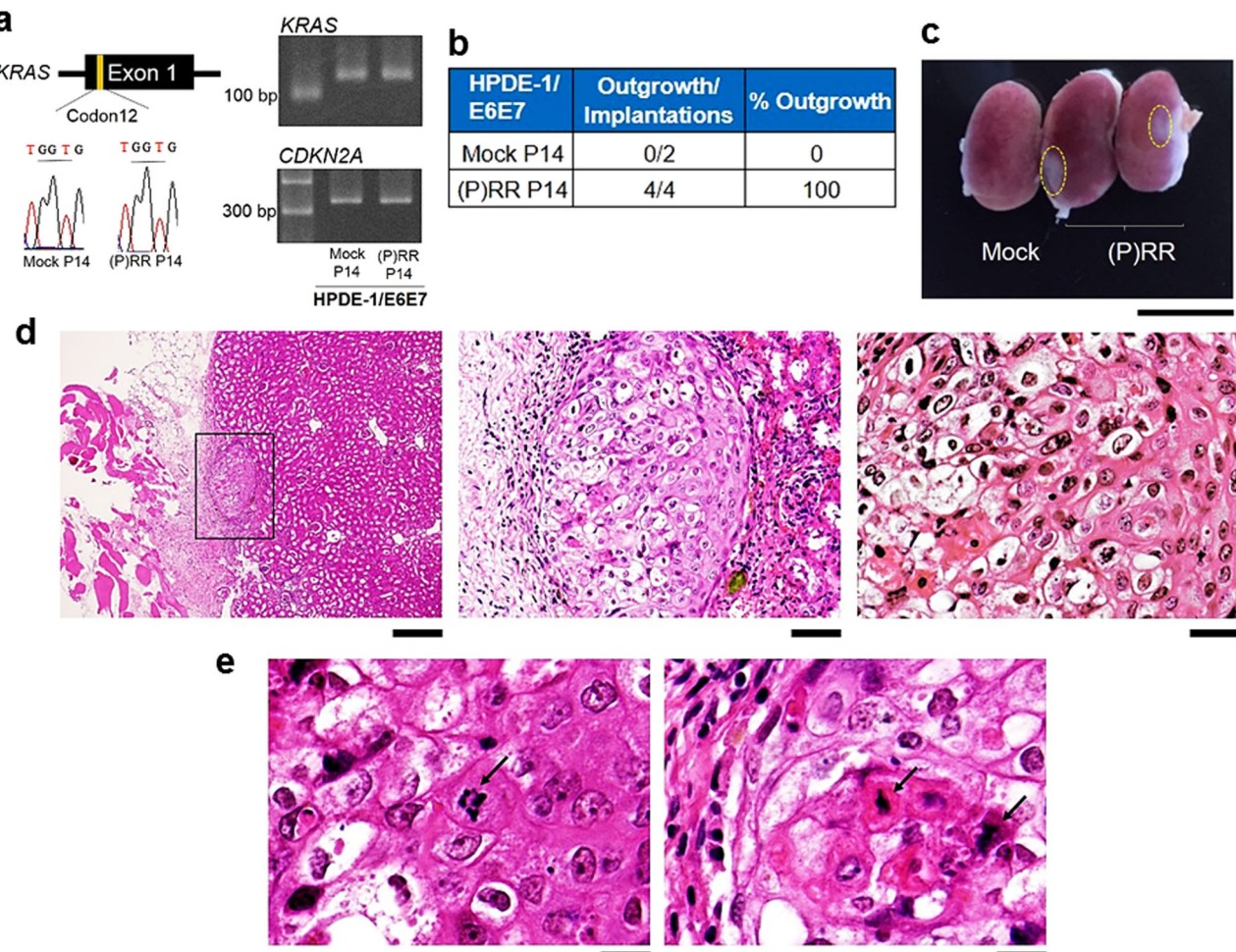

**Fig. 3 Tumour-forming ability in (P)RR-expressing HPDE-1/E6E7 cell population. a** The DNA sequence obtained by Sanger sequencing of codon 12 of *KRAS* (Left) and PCR products of *KRAS* and *CDKN2A* (Right) in 14-passage HPDE-1/E6E7 cells expressing either Mock or (P)RR. **b** Summary of the implantation and outgrowth of the HPDE-1/E6E7 cell population expressing either Mock or (P)RR. **c** Representative image showing the outgrowth of the HPDE-1/E6E7 cell population expressing (P)RR after implantation into kidney of immunodeficient mice (×2.7). Dotted line shows the area occupied by the HPDE-1/E6E7 cell populations expressing (P)RR. **d** Left: Tissue formed by (P)RR-expressing HPDE-1/E6E7 cell population engrafted into renal subcapsules of immunodeficient mice (×42). Middle: Tissue formed by the (P)RR-expressing HPDE-1/E6E7 cell population was composed of cells with different nuclear types and sizes (×200). Right: Atypical nuclei were characterized by the presence of distinctive nuclear bodies in (P)RR-expressing HPDE-1/E6E7 cells (×400). **e** Representative images indicating chromosomal abnormalities (×800), namely, the formation of a bridge between chromosomes (Left) and a fused chromosome (Right), as shown by arrows.

molecules, it was revealed that (P)RR overexpression substantially affects gene expression and DNA replication, recombination and repair (Fig. 5b and Supplementary Fig. 4d). The significant canonical pathways responsible for "Granzyme A signalling", "estrogen receptor signalling", "nucleotide excision repair (NER)" and "glucocorticoid receptor signalling" were identified by analysis of the upregulated molecules (Supplementary Fig. 4e). On the other hand, different genome stability pathways such as "cell cycle control of chromosomal replication", "base excision repair (BER)", "DNA double-strand break repair by nonhomologous end joining (NHEJ)" and "telomere extension by telomerase" pathways (Fig. 5c, d) were identified with high statistical significance under IPA in the downregulated molecules. As these pathways are marked as main pathways implicated in genomic instability[36], we also examined molecular functions composed of the downregulated molecules by using cells with the deletion of each domain of (P)RR (Fig. 5e and Supplementary Fig. 5a). Besides the inactivation of the components responsible for DNA damage response such as H2AX (Ser.139) by (P)RR overexpression (Supplementary Fig. 5b), we also confirmed significant downregulation of MCM3, PCNA, PARP1 and Ku80 encoded by *XRCC5* in both HPDE-1/E6E7 and HEK293 cells with FL(P)RR and (P)RR-△N at six-passage (Fig. 5f and Supplementary Fig. 5c). Even in 20-passage Mock cells having undergone a substantial cell division, expression of these molecules was maintained (Supplementary Fig. 5d). On the other hand, Wnt components such as pLRP6 and active β-catenin were upregulated in cells with FL(P)RR and (P)RR-△C (Fig. 5g). Cell proliferative ability was also significantly increased in these cells (N = 3 for each; one-way ANOVA; P < 0.05; Fig. 5h). Collectively, these results indicate that downregulation of molecules involved in genomic stability mediated by CTF of (P)RR is not associated with the increased number of cell division through NTF of (P)RR. We also conducted DNA fibre assay to determine the progression of individual replication forks and fork stalling. Compared to Mock-expressing HPDE-1/E6E7 cells, significantly slower fork rate and increased replication fork stalling (Mock: 0%; FL(P)RR: 19%; (P)RR-△C:0% (P)RR-△N: 20%) were detected in both FL (P)RR- and (P)RR-△N-expressing HPDE-1/E6E7 cells, which is consistent with the data from HEK293 cells with deletion mutants

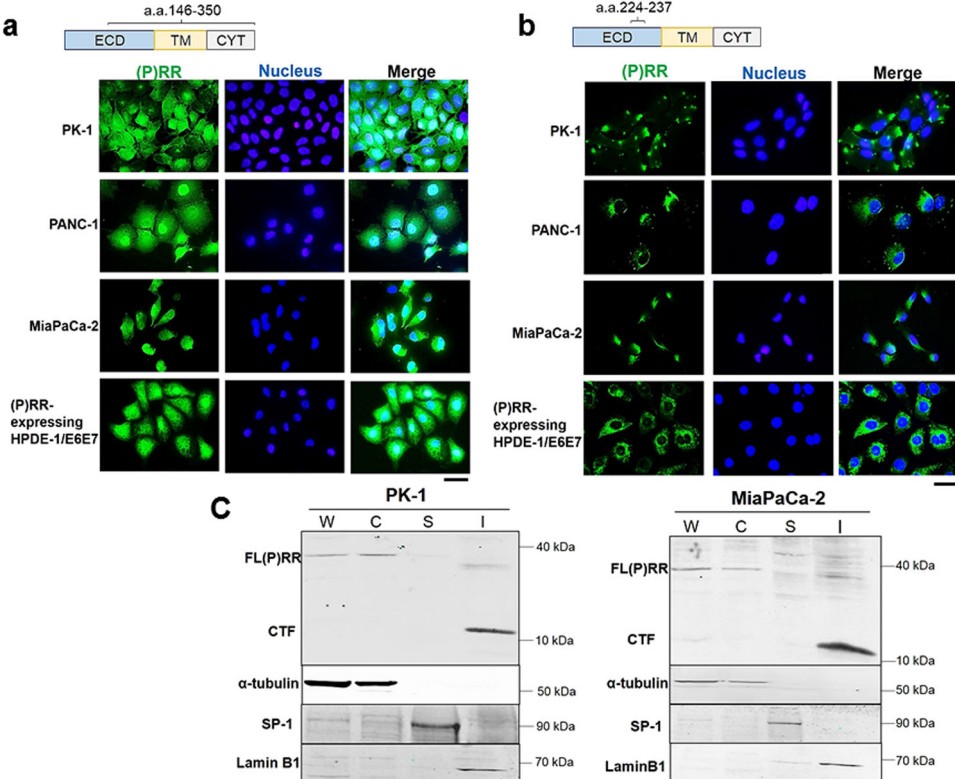

**Fig. 4 Intracellular localization of domains of (P)RR. a** (P)RR is expressed in both the cytoplasm and the nucleus in human PDAC cell lines and (P)RR-expressing HPDE-1/E6E7 cells. Immunofluorescence (IF) with anti-*ATP6AP2* antibody recognizing a.a. 146–350, which covers from the extracellular to the cytoplasmic domains (×500). ECD Extracellular domain; TM Transmembrane domain; CYT Cytoplasmic domain. **b** (P)RR is expressed in the cytoplasm, as determined by IF with anti-*ATP6AP2* antibody recognizing a.a. 224–237, which is localized in the extracellular domain (×500). **c** CTF of (P)RR is dominantly expressed in the insoluble nuclear fraction in human PDAC cell lines. Detection of each domain of (P)RR was performed using anti-*ATP6AP2* antibody recognizing a.a. 146–350. Consistent results are obtained in three independent experiments. W Whole-cell lysates; C Cytoplasmic fraction; S Soluble nuclear fraction; I Insoluble nuclear fraction.

of (P)RR ($N = 100$ for each; one-way ANOVA; $P < 0.0001$; Fig. 5i and Supplementary Fig. 6).We also determined the level of DNA damage including single- and double-strand breaks by comet assay[37]. Data showed that the percentage of distinctive DNA tails was significantly higher in cells with FL(P)RR and (P)RR-⊿N ($N = 100$ for each; one-way ANOVA; $P < 0.0001$; Fig. 5j and Supplementary Fig. 7). These results demonstrate that aberrant CTF expression of (P)RR leads to DNA replication stress and defects of DNA repair capacity. We also implemented Flow-FISH to measure the length of telomeres[38], which is considered to be an indicator of the early carcinogenesis of PDAC[39]. Data showed that increased (P)RR expression reduces telomere length by 22% ($N = 3$ for each; Mann–Whitney $U$ test; $P < 0.05$; Fig. 5k and Supplementary Fig. 8). Overall, these combined analyses revealed that aberrant expression of (P)RR leads to failure of coordination of different genomic stability pathways.

**Direct molecular binding of (P)RR with SMARCA5.** To investigate the genome integrity network responsible for transcriptional regulation, DNA replication, DNA repair and telomere maintenance, we performed a network analysis of IPA with filtered significant peptides under aberrant (P)RR expression. These analyses successfully resulted in the identification of four different functional networks with significant Ingenuity scores, which were related to molecular functions such as gene expression, cellular assembly and organization, and DNA replication, recombination and repair (Fig. 6a and Supplementary Fig. 9 and Supplementary Data 4). These molecular networks can also be

merged into a single molecular network through several common molecules (Fig. 6a and Supplementary Fig. 9 and Supplementary Data 4). Among the identified molecular networks, we focused on the network characterized by cellular assembly and organization, which also includes imitation switch (ISWI) and chromatin helicase DNA binding protein (CHD) chromatin remodelling complexes affecting histone 4 (Fig. 6a and Supplementary Data 4). Previous studies have shown that chromatin remodelling can coordinate the DNA accessibility of components involved in DNA-dependent processes by a change of the epigenomic landscape[40]. These lines of evidence, combined with the present substantial data, strongly suggest that (P)RR plays an important role in regulation of the chromatin remodelling complex. Throughout the whole molecular network, components of three different chromatin remodelling complexes, namely, ISWI, CHD and inositol 80 requiring (INO80), were also included (Fig. 6b). Moreover, SMARCA5 plays a pivotal role as an ATPase in this complex and undertakes molecular interactions with several components of other chromatin remodelling complexes[41–48], suggesting that SMARCA5 is a central player in this molecular network (Fig. 6b). In (P)RR-expressing HPDE-1/E6E7 and HEK293 cells, the upregulation of SMARCA5 was confirmed by both LC-MS/MS analysis and western blotting (Fig. 6c and Supplementary Fig. 10a). Furthermore, MCM3, PCNA, PARP1 and Ku80 as the components responsible for genome stability pathways, were downregulated in cells with aberrant SMARCA5 expression (Fig. 6d and Supplementary Fig. 10b). Similar changes were observed in cells with aberrant (P)RR expression (Fig. 5f). To explore whether (P)RR has a direct molecular interaction with

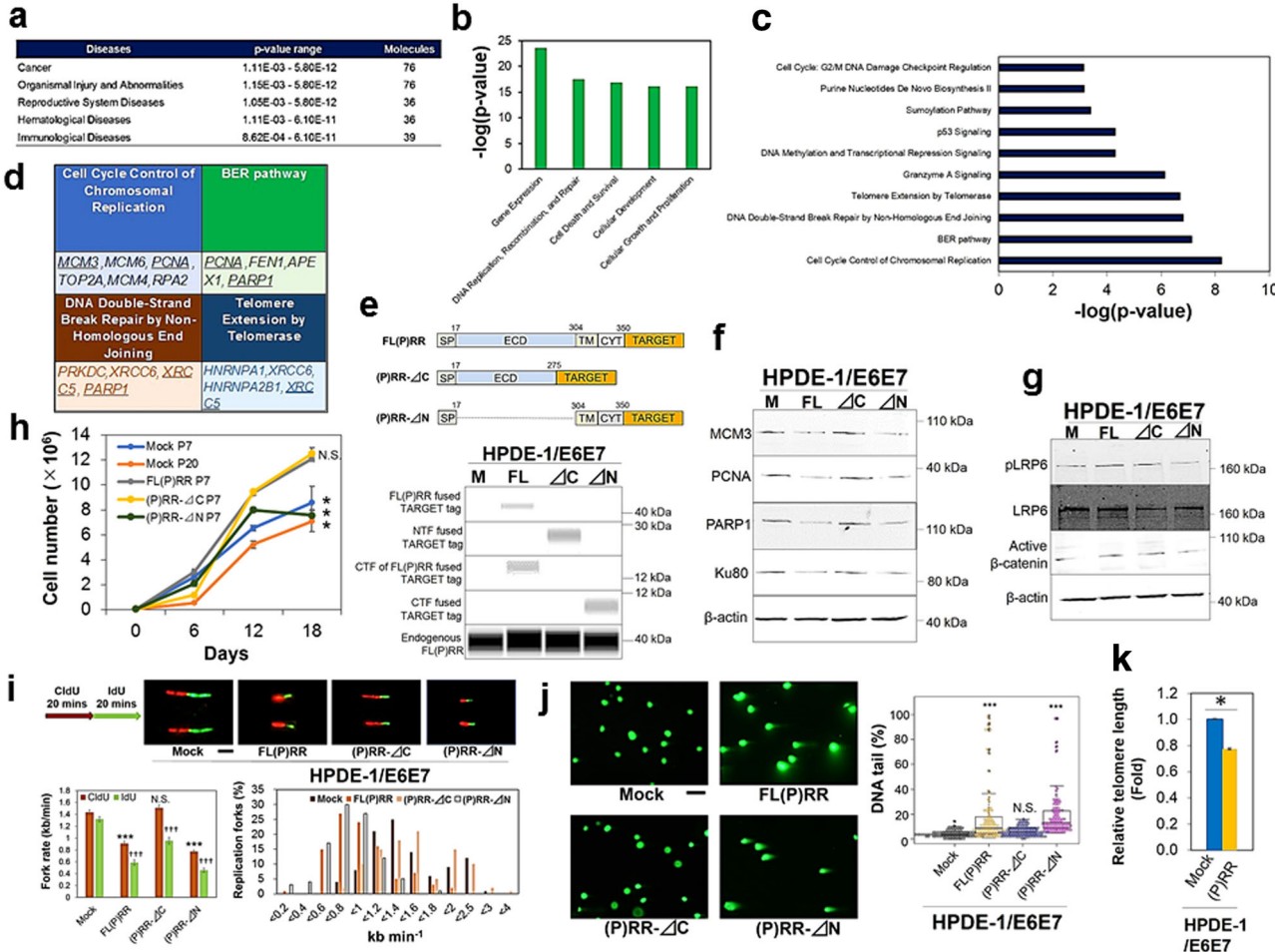

**Fig. 5 Dysfunction of genomic stability pathways by aberrant (P)RR expression in HPDE-1/E6E7 cells. a** Diseases expected from molecules downregulated by aberrant (P)RR expression. **b** Ingenuity Pathway Analysis (IPA) for molecular functions of the downregulated molecules under (P)RR overexpression. **c** Canonical pathways downregulated by (P)RR overexpression. **d** Canonical pathways identified with high IPA statistical confidence. Underline indicates molecules confirmed by western blot. **e** Constructs of deletion mutants in human (P)RR and confirmation of gene transfection in the vectors with each of Mock (M), FL(P)RR (FL), NTF of (P)RR (⊿C) and CTF of (P)RR (⊿N). Endogenous (P)RR was used as a loading control. **f** Expression of molecules involved in genomic stability pathways in cells with the deletion of each domain of (P)RR. Consistent results are obtained in three independent experiments. **g** Activation of Wnt components. Consistent results are obtained in three independent experiments. **h** Cell proliferative ability (mean ± SEM, $N = 3$ for each, *$P < 0.05$ vs. FL(P)RR, N.S., not significant). **i** DNA fibre assay. Cells were incubated sequentially with 5-chlorodeoxyuridine (CldU) and 5-iododeoxyuridine (IdU) for 20 min each. Upper: Representative images of DNA replication fork in cells (×800). Lower left: Quantitative evaluation of replication fork rates in cells (mean ± SEM, $N = 100$ for each, ***$P < 0.0001$ vs. Mock in CldU, N.S., not significant, †††$P < 0.0001$ vs. Mock in IdU). Lower right: Distribution of replication fork rate with CldU. **j** Single-cell gel electrophoresis in cells with the deletion of each domain of (P)RR at three-passage. Left: Representative images of DNA tail in cells (×100). Right: Quantification of the cells with DNA tails ($N = 100$ for each, ***$P < 0.001$ vs. Mock, N.S., not significant). The horizontal line inside the box plot is the median and the vertical lines protruding the box extend to the minimum and the maximum values, respectively. The vertical width of the central box shows the inter-quartile deviation. **k** Telomere length evaluated by Flow-FISH analyses (mean ± SEM, $N = 3$ for each, *$P < 0.05$).

SMARCA5, we performed co-immunoprecipitation using the insoluble nucleus fraction containing CTF of (P)RR. The data showed that endogenous (P)RR undertook direct molecular binding with endogenous SMARCA5 (Fig. 6e and Supplementary Fig. 10c). Treatment with two different SMARCA5 siRNAs rescued the expression of molecules responsible for genomic stability pathways in (P)RR-expressing cells (Fig. 6f and Supplementary Fig. 10d). Collectively, these results demonstrate that aberrant (P)RR expression increases SMARCA5 expression through direct molecular interaction, which results in the downregulation of several genomic stability pathways. In conclusion, our substantial data strongly indicate that aberrant (P)RR expression induces genomic instability and is an essential molecular mechanism for early carcinogenesis of PDAC (Fig. 7).

## Discussion

The present study demonstrates that aberrant (P)RR expression directly induces genomic instability and tumour-forming ability in HPDE cells. The increase of (P)RR expression also upregulates SMARCA5 through a direct molecular interaction, which results in the dysfunction of DNA maintenance, such as DNA replication, DNA repair and telomere maintenance. The data reveal that inappropriate augmentation of (P)RR expression is the fundamental molecular mechanism responsible for the early carcinogenesis of (P)RR in PDAC.

Our previous study showed that the increase of (P)RR expression synchronizes with the appearance of atypical nuclei observed in PanIN-2 of human PDAC tissues[7]. In the present study, (P)RR-expressing HPDE-1/E6E7 and HPDE-6/E6E7 cell

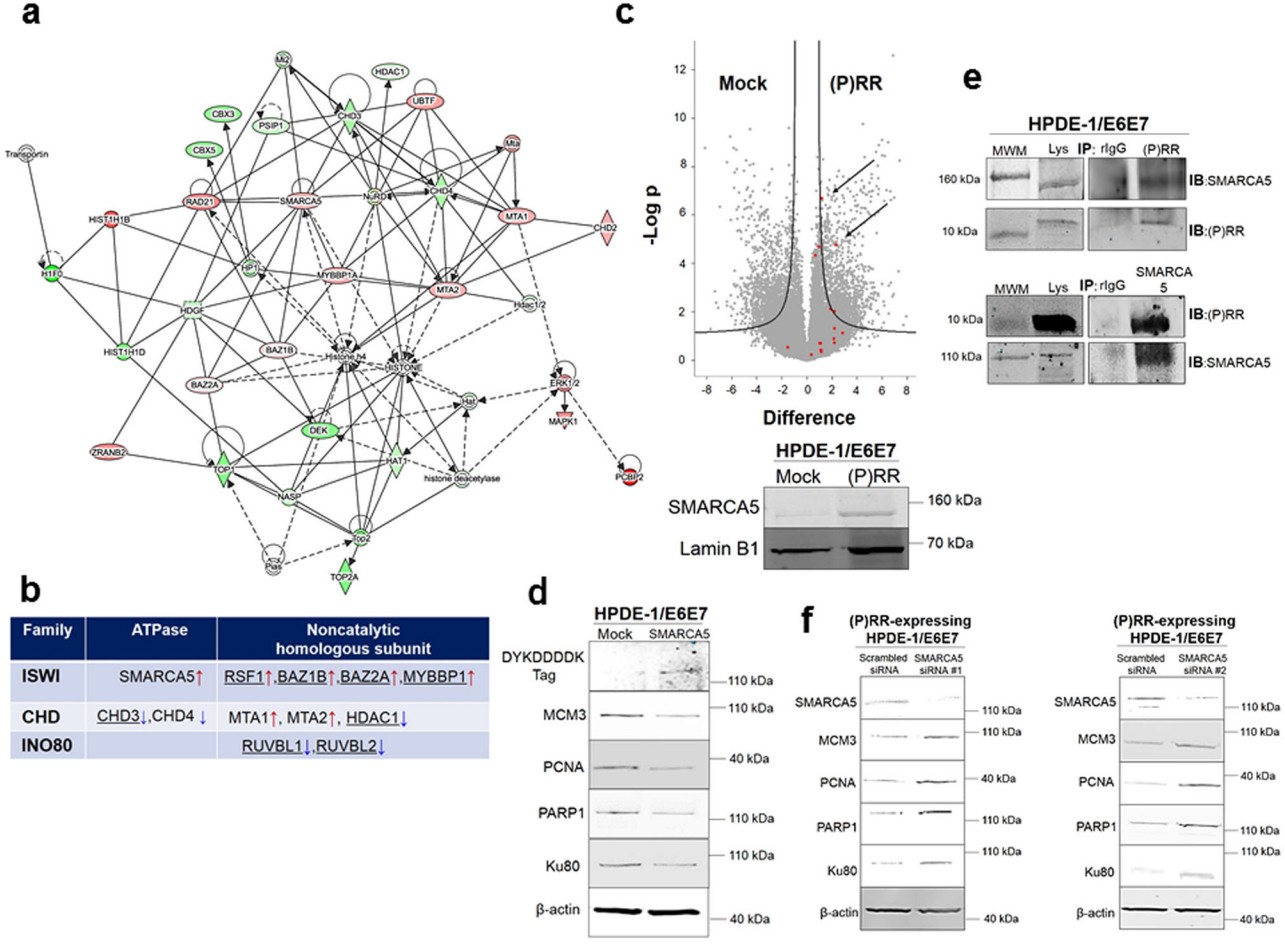

**Fig. 6 Direct molecular binding of (P)RR with SWI/SNF-related, matrix-associated, actin-dependent regulator of chromatin, subfamily a, member 5 (SMARCA5). a** Molecular interaction network affected by aberrant (P)RR expression in HPDE-1/E6E7 cell population. Red shapes: upregulated molecules; Green shapes: downregulated molecules. Straight line: direct interaction. Dotted line: indirect interaction. The components of the imitation switch (ISWI) chromatin remodelling complex are dominant in this network. **b** The components of several chromatin remodelling complexes affected by aberrant (P)RR expression. Underline: molecules interacting with SMARCA5. Red arrows: upregulated molecules; blue arrows: downregulated molecules. **c** Upper: Volcano plot of SMARCA5. Arrows show the significant upregulation of SMARCA5 under (P)RR overexpression in HPDE-1/E6E7 cells. Lower: Representative image showing the upregulation of SMARCA5 with western blot. **d** Representative image for the expression of components of DNA replication, DNA repair and telomere maintenance machinery in HPDE-1/E6E7 cells with SMARCA5 overexpression. **e** Binding of CTF of (P)RR with SMARCA5 under coimmunoprecipitation using insoluble nucleus. MWM Molecular Weight Marker; Lys Lysates; rIgG rabbit IgG; IP Immunoprecipitation; IB Immunoblot. **f** Expression of molecules responsible for genomic stability in (P)RR-expressing HPDE-1/E6E7 cells transfected with two different SMARCA5 siRNAs. Consistent results are obtained in three independent experiments for western blot and coimmunoprecipitation.

populations were mainly composed of significantly larger cells containing atypical nuclei with distinctive nuclear bodies, much variable and larger nuclear area, multinucleate as well as abnormal chromosomes, all of which are biological characters of cancer. Diverse atypical nuclei observed in (P)RR-expressing cell populations implies that future studies with genomic analysis at single-cell level will explain the detailed status of intra-tumoral heterogeneity.

Whole-genome sequencing has revealed that stable (P)RR overexpression induces large numbers of point mutations and SVs including driver genes defined by COSMIC, but does not increase indels. These results indicate that stable (P)RR overexpression induces genomic instability responsible for the generation of tumour heterogeneity. Several studies have shown that increased SVs and somatic mutations are frequently observed in PDAC[4,5]. Furthermore, SVs associated with the appearance of driver genes of PDAC and the increase of total somatic mutations were found in PanIN[5,49]. Additionally, PDAC patients with large number of SVs called as "unstable subtype" were characterized by

more aggressive behaviour, which is related to refractory[4,5]. Indeed, (P)RR expression was increased with severe stage in TNM classification in tissues of PDAC patients[6]. Taken together, these results suggest that sequential elevation of (P)RR expression contributes not only to carcinogenesis, but also the augmentation of aggressiveness and refractory representing a malignancy of PDAC patients through genetic aberration.

In the present study, (P)RR-expressing HPDE-1/E6E7 cell population exhibited tumour-forming ability. Although the explosive cell population growth was not observed, owing to the absence of *KRAS* codon 12 mutation, (P)RR-expressing cells exhibited atypical nuclei with chromosomal abnormalities of bridge and fusion generated by telomere dysfunction[31]. These findings are consistent with the concept that (P)RR-expressing HPDE-1/E6E7 cell populations undergo a pre-cancerous state and have the potential to progress towards becoming cancerous. Because theoretical analyses have shown that gene alterations accumulate in proportion to the number of cell divisions[50,51] and the level of acquired neutral mutations[52], future studies with

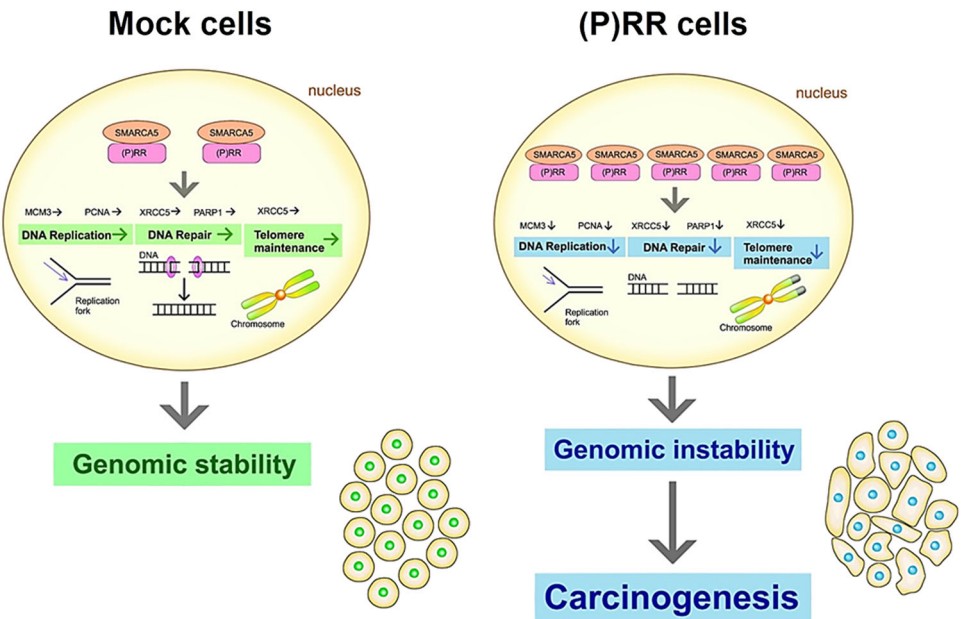

**Fig. 7 Possible molecular mechanism of genomic instability induced by aberrant (P)RR expression.** Aberrant (P)RR expression enhances SMARCA5 expression through a direct molecular interaction, which results in the failure of several genomic stability pathways. In short, aberrant (P)RR expression induces genomic instability and contributes to early carcinogenesis of PDAC.

long-term cell culture will be needed until the successful detection of driver mutations of PDAC patients in (P)RR-expressing HPDE-1/E6E7 cell population.

The present study showed that FL(P)RR and/or NTF were consistently expressed in the cytoplasm, while CTF was dominantly located in the nucleus composed of nuclear lamina and chromatin-binding protein in both human PDAC cell lines and (P)RR-expressing HPDE-1/E6E7 cells. Previous studies showed that FL(P)RR and/or NTF were localized in the cytoplasm of COS-7 and MEF cells[25,53]. Similar to human PDAC cell lines, the information from the database of The Human Protein Atlas (https://www.proteinatlas.org)[54] showed that (P)RR is expressed in both cytoplasm and nucleus in several cancer cell lines when using the antibody recognizing all the domains of (P)RR, which further supports our hypothesis that CTF of (P)RR is involved in genomic stability.

Our data reveal that aberrant (P)RR expression affects gene expression, DNA replication, recombination and repair, and induces a failure in the coordination of these genomic stability pathways. Downregulation of molecules responsible for genomic stability was observed in cells with FL(P)RR and (P)RR-△N. Furthermore, DNA fibre and comet assays indicate that CTF of (P)RR plays a critical role in the protection of DNA. In connection with the defects of DNA repair capacity, inactivation of DNA damage response leads to the failure of anti-tumourigenesis barrier function[55]. DNA replication stress is induced by the reduction of minichromosome maintenance complex component (MCM)2-7 constituting the core of the replicative DNA helicase[56]. Thus, it is possible that the increase in DNA replication stress is involved in the downregulation of components of MCM complex induced by aberrant (P)RR expression. It has also been shown that DNA replication stress leads to the accumulation of DNA lesions and induces striking chromosomal instability in cancer cells in the absence of mitotic dysfunction[57]. Interestingly, our data have revealed that aberrant (P)RR expression induces DNA replication stress and SVs including several dozens of large deletions, suggesting DNA replication stress is not due to mitotic dysfunction. Namely, domain of (P)RR regulating genomic stability including DNA replication is mediated by CTF of (P)RR,

which is independent of NTF of (P)RR induced cell proliferation. The scarcity of indels observed in the mutational landscape of stable (P)RR-overexpressing cells also supports the concept that DNA replication stress is not mediated by cell proliferation. Since loss of function of BER enhances the somatic mutation rate[58], massive SVs and elevated rate of somatic mutations detected by whole-genome sequencing will be the consequence of the above genetic change induced by (P)RR overexpression.

Dysfunction of NHEJ leads to failure of regulation of the correct length of telomeres and of protection of their ends[59]. In association with the reduction of Ku80 expression[60], shortened telomeres are observed as an initial genetic change, which are found in more than 90% of PDAC tissues[39,61,62]. Consistent with these pathological analyses in PDAC patients, the present study reveals that aberrant (P)RR expression reduces the telomere length associated with the downregulation of *XRCC 5* encoding Ku80, which is associated with complex SVs. These findings support our hypothesis that an increase of (P)RR expression is an essential molecular mechanism behind the early carcinogenesis in PDAC. In this context, our data previously indicated that plasma-soluble (P)RR is a potential biomarker to identify patients with PDAC from an early stage[7].

We have detected that aberrant (P)RR expression upregulates SMARCA5, which is the catalytic subunit of ATPase, to form ISWI chromatin remodelling complex coordinating the access of DNA binding proteins to DNA[63]. By using information from The Human Protein Atlas[54], we have confirmed that SMARCA5 is aberrantly expressed in over 80% of PDAC patients. In SMARCA5-expressing cells, several components of the DNA replication, DNA repair and telomere maintenance machinery were markedly downregulated, which coincides with the results of (P)RR-expressing cells. In the present study, we have demonstrated a direct molecular binding between (P)RR and SMARCA5. Furthermore, reduction in SMARCA5 rescued the expression of molecules responsible for genomic stability in (P)RR-expressing cells. These data support the hypothesis that the upregulation of SMARCA5 is induced by aberrant (P)RR expression through their direct molecular interaction, which plays an important role in generating genetic evolution of PDAC. Our

data also showed that (P)RR overexpression reduces PARP1 expression through the upregulation of SMARCA5. To accomplish DNA damage responses followed by the NHEJ repair pathway, PARP1 poly ADP-ribosylates (PARylates) histone tails, resulting in the promotion of chromatin relaxation and nucleosome eviction from DNA[64,65]. These processes allow the recruitment of several chromatin remodellers, such as SMARCA5 and CHD4, through the binding to PAR, which further relaxes chromatin to facilitate DNA repair, such as NHEJ. Thus, reduction in PARP1 expression induced by SMARCA5 overexpression may prevent not only the relaxation of chromatin, but also the recruitment of SMARCA5, followed by limiting the access of DNA repair proteins to chromatin.

Although the clear genomic instability is intrinsically linked to significant alterations in apoptosis control[66], cell proliferative ability was not affected in (P)RR-expressing cells. It has been shown that NTF of (P)RR is actually a component of the Wnt receptor complex[12]. The data from the present study also indicate that NTF of (P)RR contributes to cell proliferation through the activation of Wnt/β-catenin signalling pathway. Other investigators have also indicated that Wnt signalling in cooperation with mitogen-activated protein kinase activation promotes the proliferation of established PanIN lesions[67]. Furthermore, it has also been shown that aberrant (P)RR expression contributes to activation of the PI3K/Akt signalling pathway[6]. Collectively, these findings suggest that the orchestrated activities of different signalling pathways promote maintenance of the proliferative ability of (P)RR-expressing cells, despite marked genomic instability.

In conclusion, the present study demonstrates that aberrant (P)RR expression induces genomic instability by an increase in SMARCA5 expression through a direct molecular interaction. These data indicate that inappropriate augmentation of (P)RR expression contributes to the evolution of PDAC.

## Methods

**Cell lines**. Immortalized HPDE-1/E6E7[23] and HPDE-6/E6E7[24] cells were cultured in Hu-Media KG2 (Kurabo, Osaka, Japan; catalogue #KK-2150S), which were gifted from Dr. Furukawa. In addition, HEK293 cells were cultured in DMEM (Sigma-Aldrich, St. Louis, MO, USA; catalogue #D5796) supplemented with 10% FBS, penicillin and streptomycin. Both media containing 400 μg/mL G418 (Sigma-Aldrich; catalogue #A1720) as an antibiotic were used to culture successfully transfected cells. PK-1, PANC-1 and MiaPaCa-2 of human PDAC cell lines and Human T-cell leukaemia 1301 cells were cultured in RPMI-1640 (Sigma-Aldrich; catalogue #R8758) supplemented with 10% FBS, penicillin and streptomycin. These cell lines were maintained at 37 °C under 5% $CO_2$/95% air in a humidified incubator. HEK293 was drawn from stocks at Department of Pharmacology, Faculty of Medicine, Kagawa University. PK-1 was originally from Institute of Development, Aging and Cancer (IDAC). PANC-1 and MiaPaCa-2 were purchased from American Type Culture Collection (ATCC). 1301 were from European Collection of Cell Cultures (ECACC). The sources of cell lines were also listed in Supplementary Data 5. All the cell lines used in this study have been tested for mycoplasma by MicoAlert (Lonza, Tokyo, Japan; catalogue #BWLT07). No commonly misidentified lines were used in the study.

**Vector construction**. pOTB7 vector inserted with human ATP6ap2 was obtained from GE Healthcare (Chicago, IL, USA; catalogue #6575732). A DNA fragment corresponding to human ATP6ap2 (1050 bp without a stop codon) encoding (P)RR was inserted into the restriction sites of BamHI and AgeI within the mammalian expression plasmid pEB Multi-Neo TARGET tag-C (Wako, Osaka, Japan; catalogue #165-26521) to generate stable (P)RR-expressing HPDE-1/E6E7, HPDE-6/E6E7 and HEK293 cells. Human ATP6ap2 inserted into pcDNA 3.0 tagged with 10His[25] was also used for making stable and non-replicative transient (P)RR-expressing cells. Plasmid pEB Multi-Neo TARGET tag-C were used to construct not only human (P)RR with the full length [FL (P)RR], but also ⊿NTF of (P)RR [amino acids (a.a.) 304–350; (P)RR-⊿N] and ⊿CTF of (P)RR [a.a. 17–275; (P)RR-⊿C]. The sequence integrity of all inserts was confirmed by Sanger DNA sequencing. The open reading frame of human SMARCA5 tagged with Myc-DDK (RC203775) was also purchased from ORIGENE (Rockville, MD, USA).

**Plasmid and siRNA transfection**. In accordance with the manufacturer's instructions, FL(P)RR-, (P)RR-⊿N-, (P)RR-⊿C- or SMARCA5-expressing vector or an empty vector was transfected into HPDE-1/E6E7, HPDE-6/E6E7 and

HEK293 cells using Lipofectamine 3000 (Thermo Fisher Scientific; catalogue #L3000008). Using Lipofectamine RNAiMAX (Thermo Fisher Scientific; catalogue #13778075), SMARCA5 Stealth RNAi™ siRNA was transfected into (P)RR-expressing cells according to the recommended protocol. The sequences used to generate SMARCA5 Stealth RNAi™ siRNA #1 were as follows: 5′-GGA GGC UUG UGG AUC AGA AUC UGA A-3′and 5′-UUC AGA UUC UGA UCC ACA AGC CUC C-3′. The sequences used to generate SMARCA5 Stealth RNAi™ siRNA #2 were as follows: 5′-CAG GGA AGC UCU UCG UGU UAG UGA A-3′ and 5′-UUC ACU AAC ACG AAG AGC UUC CCU G-3′. For the scrambled siRNA, Stealth RNAi™ siRNA Negative Control Med GC (Thermo Fisher Scientific; catalogue #12935-300) was used.

**Confirmation of gene transfection**. (P)RR fused to Target tag (168 bp) and to 10His tag (244 bp) was detected by PCR. The primer sequences used are listed in Supplementary Data 5. The PCR was performed using Takara Ex Taq (Takara Bio; catalogue #RR001A). PCR conditions were as follows: 94 °C for 2 min, followed by 40 cycles of denaturation at 98 °C for 10 s, annealing at 64 °C for 30 s and extension at 72 °C for 1 min. The final extension was then performed at 72 °C for 5 min. PCR products were visualized by gel electrophoresis. The detection of FL(P)RR, (P)RR-⊿N and (P)RR-⊿C was confirmed using Simple Western™ System (ProteinSimple, San Jose, CA, USA), in accordance with previous reports[68] and the manufacturer's instructions. The protein concentration in each sample was set as 2 μg/μL and was serially diluted to detect the target band. Separation electrophoresis and immunodetection were performed automatically. The detection of deletions of the (P)RR domain was performed using anti-ATP6ap2 antibody having an immunogen corresponding to a region within a.a.146 and 350 (Sigma-Aldrich; catalogue #SAB2702080).

**Cellular atypia**. We evaluated the cell area of HPDE-1/E6E7 and HPDE-6/E6E7 cells expressing Mock or (P)RR using Image J. Papanicolaou stain was also performed. Multinucleated cells were evaluated by visual observation and the nuclear area was measured in Image J using Mock- and (P)RR-expressing HPDE-1/E6E7and HPDE-6/E6E7 cells.

**In vivo transplantation**. Approval for this experiment was obtained from the Animal Experimentation Ethics Committee of Kagawa University. Five-week-old male BALB/c immunodeficient mice (nu+/nu+) (CLEA, Tokyo, Japan) were used for the implantation experiments. Three-dimensional spheroids were prepared from HPDE-1/E6E7 cells expressing either Mock or (P)RR, in accordance with a previous report[69]. In all, $5 \times 10^3$ cells from each group were seeded onto a V-bottomed 96-well plate (Sumitomo Bakelite Co. Ltd., Tokyo, Japan; catalogue #MS-9096V) with 150 μL of culture medium in each well and cultured for four days. Then, 20 aggregates prepared from HPDE-1/E6E7 cells expressing either Mock or (P)RR were implanted into the renal subcapsules of immunodeficient mice. Five weeks later, these animals were sacrificed and the kidney sections including implanted spheroids were collected. After embedding in paraffin, the sectioned samples were stained with haematoxylin and eosin.

**Direct cell counting**. We performed a count of the total number of HPDE-1/E6E7 cells expressing either Mock or the deletion of each domain of (P)RR over 18 days. The cells were treated with Accutase (Merck Millipore, Billerica, MA, USA; catalogue #SCR005). They were then re-suspended in PBS after centrifugation. Cell suspension was stained with 0.4% trypan blue stain solution to evaluate the number of living cells. Total cell number per millilitre was measured using a Countess® Automated Cell Counter (Thermo Fisher Scientific; catalogue #C10227).

**DNA extraction**. To collect DNA, we prepared HPDE-1/E6E7 cells expressing either Mock or (P)RR. The genomic DNA from each cell was then extracted using the QIAGEN Blood & Cell Culture Kit (Qiagen; catalogue #13362).

**PCR and DNA sequencing for KRAS and CDKN2A**. For the detection of somatic mutations in KRAS and CDKN2A, primers were used for PCR amplification, as described previously[70]. The primer sequences are listed in Supplementary Data 5. Using Takara Ex Taq, initial denaturation was performed at 94 °C for 2 min, followed by 30 cycles of denaturation at 98 °C for 10 s, annealing at 52 °C for KRAS and at 65 °C for CDKN2A for 30 s, and extension at 72 °C for 1 min. Then, the final extension was performed at 72 °C for 5 min. PCR products were visualized by gel electrophoresis. BigDye Terminator v3.1 Cycle Sequencing Kit (Thermo Fisher Scientific; catalogue #4336917) and BigDye Terminator Purification Kit (Thermo Fisher Scientific; catalogue #4376484) were used to sequence the PCR products. The DNA sequence was evaluated by ABI PRISM 3100 Genetic Analyzer (Thermo Fisher Scientific), based on the manuals provided by the manufacturer.

**Immunofluorescence of (P)RR**. PK-1, PANC-1, MiaPaCa-2 cells and (P)RR-expressing HPDE-1/E6E7 cells were grown on chamber slides (Matsunami, Osaka, Japan; catalogue #SCS-002). The cells were fixed in 4% paraformaldehyde phosphate-buffered solution (Wako; catalogue #163-20145) at room temperature for 5 min. After washing with PBS containing 0.1% Tween 20, the cells were then

incubated for 20 min in blocking solution containing fetal bovine serum. The cells were incubated at 4 °C overnight with either Anti-*ATP6ap2* antibody produced in a rabbit as an immunogen corresponding to the region within a.a. 146 and 350 (Sigma-Aldrich; catalogue #SAB2702080) or a.a. 224 and 237[33]. After washing with PBS containing 0.1% Tween 20, cells were incubated at room temperature for 2 h with Goat anti-rabbit IgG (H + L) highly cross-adsorbed secondary antibody, Alexa Fluor anti-rabbit 488 (Thermo Fisher Scientific; catalogue #A11034), to detect (P)RR expression labelled by different Anti-*ATP6ap2* antibodies. The nuclei were stained by H33258 (Sigma-Aldrich; catalogue #B2883). Fluorescence images were obtained using Olympus FSX 100 (Olympus, Tokyo, Japan).

**Western blot analysis**. Measurement of protein concentration after lysis of cells and protein extraction were performed by Bradford protein assay[7]. Total protein extracts (30 µg) were electrophoretically separated using polyacrylamide gels with SDS ranging from 10% to 14% and transferred onto nitrocellulose membranes. For protein expression, we used blocking buffer and secondary antibodies coupled to infrared dyes required for the Odyssey scanner (LI-COR, Lincoln, NE, USA)[7]. The period for incubation is 1 h at room temperature in each process. The primary and secondary antibodies used are described in Supplementary Data 5.

**Protein extraction from insoluble nuclear fraction**. LysoPure™ Nuclear and Cytoplasmic Extractor Kit (Wako; catalogue #295-73901) allowed us to extract cytoplasmic, soluble and insoluble nuclear fractions to elucidate the intracellular localization of domains of (P)RR. In accordance with the manufacturer's instructions, the insoluble nuclear fraction was extracted using SDS lysis buffer and sonication after the extraction from cytoplasmic and soluble nuclear fractions.

**DNA fibre assay**. To evaluate the DNA replication stress upon aberrant (P)RR expression, we performed DNA fibre assay. Cells were pulse-labelled with 100 µM CldU (Sigma-Aldrich; catalogue #C6891) for 20 min, and then with 100 µM IdU (Tokyo Chemical Industry, Tokyo, Japan; catalogue #I0258) for 20 min. After cells had been lysed, the cell suspension was spread on a slide glass. Once dried, the DNA spread was fixed in a 3:1 solution of methanol-acetic acid, pre-chilled 70% ethanol and methanol. After washing with PBS, DNA was denatured with 2.5 N HCl for 30 min. Samples were rinsed with PBS and incubated with the following antibodies: anti-BrdU rat IgG and anti-BrdU mouse IgG in 1% BSA. The details of the primary antibodies are described in Supplementary Data 5. After incubation for 1 h at room temperature, the slides were washed three times for 3 min in PBS containing 0.05% Tween 20. The slides were incubated with the following secondary fluorescent antibodies: Alexa anti-mouse 488 (Thermo Fisher Scientific; catalogue #A11001) and Alexa-anti rat 594 (Thermo Fisher Scientific; catalogue #A11007) for 1 h at room temperature. After washing with PBS containing 0.05% Tween 20, slides were mounted using Fluoromount™ (Sigma-Aldrich; catalogue #C4680). Signals were measured using Image J.

**Detection of DNA damage**. The DNA repair capacity upon aberrant (P)RR expression was measured by comet assay[37]. For this assay, the OxiSelect™ Comet Assay Kit (Cell Biolabs, Inc., San Diego, CA, USA; catalogue #STA-350) was used. The "tail" resulting from the damaged DNA was separated from intact DNA by single-cell gel electrophoresis. Cells were resuspended at $1 \times 10^5$ cells/mL in ice-cold PBS. Cell suspension was combined with Comet Agarose at a 1:10 ratio (v/v) and loaded onto the OxiSelect™ Comet Slide. Under dark conditions, embedded cells were treated for 30 min at 4 °C in lysis buffer and then an alkaline solution to relax and denature the DNA. Samples were electrophoresed to detect single-and double-stranded DNA damage by using alkaline electrophoresis buffer. After rinsing the slides with distilled water, they were immersed in 70% ethanol. DNA was stained with Vista Green DNA Dye. Tail DNA intensity (%) was calculated by OpenComet[71].

**Telomere length measurement by Flow-FISH**. Telomere length was measured using Telomere PNA Kit/FITC (DAKO, Glostrup, Denmark; catalogue #K5327). Telomere length of the HPDE-1/E6E7 cells expressing either Mock or (P)RR was determined. Human T-cell leukaemia 1301 cells (ECACC, Salisbury, UK; catalogue #EC01051619-G0) characterised by tetraploidy and long telomeres were used as an internal standard. Test and control cells were mixed at a ratio of 1:1 and a total of $4 \times 10^6$ cells was resuspended in 300 µL of hybridization solution with or without a fluorescein isothiocyanate (FITC)-conjugated telomere probe. DNA was denatured for 10 min at 82 °C. In situ hybridization was performed at room temperature under dark conditions. Cells were rinsed with washing solution and heated for 10 min at 40 °C. They were then resuspended in 0.5 mL of DNA staining solution containing propidium iodide and RNase A and incubated for 3 h at 4 °C in the dark. Samples were analysed by flow cytometry. FlowJo (Tomy Digital Biology Co., Ltd., Tokyo, Japan) was used for analysis and relative telomere length (RTL) was calculated based on mean fluorescence intensity (MFI)[72].

**Coimmunoprecipitation**. For coimmunoprecipitation, we used the insoluble nuclear fractions in HEK293 and HPDE-1/E6E7 cells. A total of 50 µL of pre-washed Dynabeads Protein G (Thermo Fisher Scientific; catalogue #10003D) was coupled with 10 µg of anti-h(P)RR antibodies, anti-SMARCA5 antibodies or

nonspecific rabbit IgG diluted in 200 µL of PBS with Tween-20 for 1 h at room temperature under the rotator. The information for antibodies used are described in Supplementary Data 5. One milligram of protein was incubated with antibody conjugated with Dynabeads for 2 h at 4 °C. For immunoblotting, equal amounts of immunocomplexes or lysates were utilized and resolved by SDS-PAGE.

**Whole-genome sequencing and mutation calling**. In six-passage HPDE-1/E6E7 cells expressing stable Mock and (P)RR using pEB Multi-Neo TARGET tag-C, DNA was also quantified using the Quant-iT dsDNA BR Assay Kit (Thermo Fisher Scientific; catalogue #Q33120) to produce a whole-genome library in accordance with manufacturer's recommendations. For PCR-Free libraries, the TruSeq DNA PCR-Free LT Library Prep Kit (Illumina, San Diego, CA, USA; catalogue #FC-121-9006DOC) was used. In accordance with the manufacturer's instructions (Part #15036187 Rev. D, June 2015), we prepared whole-genome libraries as described below. DNA (1 µg) was diluted to 20 ng/µL using resuspension buffer before DNA fragmentation. DNA was fragmented to ~350 bp using Acoustic solubilizer (Covaris, Woburn, MA, USA). DNA fragment ends were repaired using End Repair Mix2. Following the end repair of DNA fragments, we performed size selection using Agencourt AMPure XP (Beckman Coulter, catalogue #A63880) to select an appropriate library size. A single "A" nucleotide was added to the 3′ ends of the blunt fragments to prevent them from ligating with each other during the process of adaptor ligation and size-selected libraries were then ligated with indexed adaptors. The quality of the final whole-genome libraries was evaluated using the Agilent 2100 BioAnalyzer (G2940CA).

In accordance with the cBot User Guide Rev. L (Part #15006165), whole-genome libraries were prepared for cluster generation by cBot. To analyse sequences on the next-generation sequencer, the flow cells were clustered on the cBot using HiSeq PE Cluster Kit v4 cBot (Illumina; catalogue #PE-401-4001). DNA libraries were analysed on the Illumina HiSeq 2500 instrument using HiSeq SBS Kit v4-H (catalogue #FC-401-4002) to perform paired-end 125-bp sequencing. In HPDE-1/E6E7 cells expressing transient Mock and (P)RR and HPDE-1/E6E7 cells without transfection, libraries were prepared using the TruSeq™ DNA PCR-Free Library Preparation Kit (Illumina; catalogue #FC-121-3001). Cluster generation and sequencing were prepared on a Novaseq 6000 system using Novaseq 6000 S2 Reagents (Illumina; catalogue #200012860). Paired-end 150-bp sequencing was performed. The coverage was set to 30x for all the samples.

Read sequences were mapped by Burrows-Wheeler Aligner (BWA) to the human reference genome (GRCh37)[73]. Possible PCR duplicated reads were removed using SAMtools and an in-house program[74]. Point mutations and short indels were identified with variant allele frequency using MCV pipeline[26]. Structural variations (SVs) were detected by mapping information. Read pairs with inconsistent mapping patterns were identified, and SVs supported by ≥4 inconsistent read pairs were called. Possible mapping errors were removed based on re-alignment. The details of the mutation calling methods are described in Fujimoto et al.[26]. We used HPDE-1/E6E7 cells without transfection as the reference and mutations were detected by comparing each Mock- and (P)RR-overexpressing HPDE-1/E6E7 cells against the reference. Since mutations should have accumulated in HPDE-1/E6E7 cells before transfection, we removed common mutations detected in 2 or more samples.

**Protein identification using nano LC-MS/MS**. We performed nanoscale liquid chromatography with tandem mass spectrometry (LC-MS/MS) to elucidate the differences of whole protein expression between Mock- and (P)RR-expressing HPDE-1/E6E7 cells. Pierce™ Mass Spec Sample Prep Kit for Cultured Cells (Thermo Fisher Scientific; catalogue #84840) allowed us to perform the reduction, alkylation, acetone precipitation and trypsin digestion using proteins extracted from the insoluble nuclear fraction of Mock and (P)RR-expressing HPDE-1/E6E7 cells, respectively. The peptides were loaded onto the LC system (EASY-nLC 1000; Thermo Fisher Scientific, San Jose, CA, USA) equipped with a trap column (Acclaim PepMap 100 C18 LC column, 3 µm, 75 µm ID × 20 mm; Thermo Fisher Scientific) equilibrated with 0.1% formic acid and eluted with a linear acetonitrile gradient (0%–35%) at a flow rate of 300 nL/min. The eluted peptides were separated on the column (EASY-Spray C18 LC column, 3 µm, 75 µm ID × 150 mm; Thermo Fisher Scientific) with a spray voltage of 2 kV (ion transfer tube temperature: 275 °C). By using MS (Orbitrap Fusion ETD MS; Thermo Fisher Scientific), the peptide ions were detected in the data-dependent acquisition mode with the installed Xcalibur software (version 4.0; Thermo Fisher Scientific). Full-scan mass spectra were acquired in the MS over 375–1500 *m/z* at a resolution of 120,000. The most intense precursor ions were selected for collision-induced fragmentation in the linear ion trap under normalized collision energy of 35%. Dynamic exclusion was employed within 60 s to prevent repetitive selection of peptides.

**MS data analysis**. The MS/MS searches were carried out using MASCOT (Version 2.6.1; Matrix Science, London, U.K.) and SEQUEST HT search algorithms against the SwissProt Homo sapiens protein database (2017-05) using Proteome Discoverer (PD) 2.1 (Version 2.1.1.21; Thermo Scientific). The search algorithms in this study were spectrum selector, Mascot, SEQUEST HT search nodes, percolator and ptmRS nodes. Carbamidomethyl cysteine was set as a fixed modification and oxidation of methionine, carbamidomethylation of histidine, lysine, peptide

N-terminus and acetylation of protein N-terminus were considered as variable modifications. MS and MS/MS mass tolerances were set to 10 ppm and 0.6 Da, respectively. Trypsin was specified as the protease and a maximum of two missed cleavages were allowed. The target-decoy database search results used for calculation of the false discovery rate (FDR) was set at 0.01.

**Bioinformatic analysis.** Label-free quantification (LFQ) was also processed with PD2.1 using precursor ion area detector mode. For differential analysis of the relative abundances of peptides between samples, Perseus[34] was used. The results were filtered using the following criteria. We utilized the peptides including three valid values in at least one group to ensure reliability of the data. Normalization of intensities was performed by subtracting the median of each sample and missing values were imputed according to a normal distribution (width = 0.3, down-shift = 1.8).

A heat map based on LFQ was clustered using Euclidean distance to confirm the suitability of data analysis. A volcano plot was also depicted by a global permutation-based FDR approach implemented in Perseus. The level of significant cut-off value is determined based on a permutation-based FDR of 0.05 and S0 value of 0.5 as a minimal fold change. For pairwise comparison of groups, two-sample Student's t-test (two-sided) was used. Ingenuity Pathway Analysis (IPA) was also utilized to evaluate the enrichment of functional analysis, canonical pathway and molecular network categories affected by aberrant (P)RR expression. Considering the evidence that CTF of (P)RR is expressed in the insoluble nuclear fraction, including chromatin-binding protein, significantly identified peptides that are also characterized by "chromatin" in GOCC (Gene Ontology Cellular Components) and/or "DNA-binding" in Gene Ontology Molecular Function (GOMF) were overlain onto the updated information developed from the Ingenuity Pathways Knowledge Base.

**Statistics and reproducibility.** At least two cell lines were used for several techniques, such as Papanicolaou stain, immunohistochemistry, western blot, DNA fibre assay, comet assay and coimmunoprecipitation to show reproducibility. We conducted these experiments to confirm data of HPDE-1/E6E7 cells presented in WGS and LC-MS/MS. Regarding the in vivo experiments, four of the kidneys were injected with (P)RR-expressing cells. All of the tissues formed by the injected (P)RR-expressing cells were composed of atypical cells. Results are expressed as mean ± SEM and beeswarm. Beeswarm was depicted by R (The R Foundation for Statistical Computing Platform). The Student's t-test was used for the analyses of cell and nuclear area. Telomere lengths in Flow-FISH was tested by Mann–Whitney $U$ test. We also used one-way ANOVA with Scheffe's post hoc test to analyse the replication fork rates in the DNA fibre assay and DNA tails in the comet assay and with Tukey-Kramer's post hoc test to analyse a cell proliferative ability in the deletion of each domain of (P)RR. Based on one hundred biological replicates in cell and nuclear area, DNA fibre assay and comet assay, cutoff was regarded as $P < 0.0001$. $P < 0.05$ was adopted in triplicate direct cell counting and Flow-FISH. The statistics are indicated in the specific "Results" and associated figure legends. All statistical analyses were performed using IBM SPSS Statistics 20 software (Armonk, NY, USA).

**Reporting summary.** Further information on research design is available in the Nature Research Reporting Summary linked to this article.

## Data availability

All processed data and analyses generated during the current study are included in this article and its Supplementary Information files, including source data: Supplementary Data 1–5. For WGS data, Fastq files were deposited with the National BioScience Database Center (NBDC) as file accession No. JGAS00000000143. For MS data, RAW data, peak lists and result files have been deposited as accession No. PXD 010107 with the ProteomeXchange Consortium[75] via the jPOST[76] partner repository as JPST000440. All the data produced in this study are stored at Department of Pharmacology, Faculty of Medicine, Kagawa University and available from the corresponding author upon reasonable request.

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

## Acknowledgements

We are grateful to Mr Kouichi Yube (Division of Research Instrument and Equipment, Kagawa University) for handling Flow Cytometry and ABI PRISM 3100 Genetic Analyzer, Mr Toru Matsunaga (Department of pathology, Kagawa University Hospital) for performing Papanicolaou stain, Ms Miho Seki (Department of Pharmacology, Faculty of Medicine, Kagawa University) for help with the western blotting and agarose gel electrophoresis, and Mr Yuji Yokota (Department of Pharmacology, Faculty of Medicine, Kagawa University) for taking care of the immunodeficient mice. Mr Koichi Ishiguro (AXIOHELIX Co. Ltd., Okinawa, Japan) also helped to perform the analysis of the human whole-genome sequence. Ms Kanako Ohtsuki created the images of Fig. 7. We also thank Edanz Group (www.edanzediting.com/ac) for editing a draft of this manuscript. This study was supported by Grants-in-Aid for Scientific Research (16K14610 and 19K07690 to Y.S. and 18H03191 to A.N.) from the Ministry of Education, Science and Culture of Japan and from the Tokyo Biomarker Innovation Research Associate (TOBIRA), the Hoansha Foundation and Alumni Association of Kagawa University (Sanjukai).

## Author contributions

Y.S., H.Y., D.Y., A.R., Ta.F., Y.F., S.T. and H.O. performed the experiments. Y.S., J.Y., S.Y., A.F. and A.N. designed this study. Y.S., H.Y., J.Y., J.W., S.Y., A.F. and A.N. wrote the paper. Y.S., K.T., H.Y., J.Y., A.F., H.O. and A.N. performed the analyses and interpretation of data. K.T., J.Y., H.Ko., T.M., Y.Y., H.Ki., S.Y., To.F. and T.N. contributed to materials/analytical tools.

## Competing interests

The authors declare no competing interests.

## Additional information

 COMMUNICATIONS BIOLOGY | https://doi.org/10.1038/s42003-020-01434-x

Yuki Shibayama[1], Kazuo Takahashi[2], Hisateru Yamaguchi [3,4], Jun Yasuda[5,6], Daisuke Yamazaki[1], Asadur Rahman [1], Takayuki Fujimori[7,8], Yoshihide Fujisawa[9], Shinji Takai[10], Toru Furukawa [11], Tsutomu Nakagawa[12], Hiroyuki Ohsaki[13], Hideki Kobara[7], Jing Hao Wong[14], Tsutomu Masaki[7], Yukio Yuzawa[2], Hideyasu Kiyomoto[15], Shinichi Yachida[16], Akihiro Fujimoto[14✉] & Akira Nishiyama [1✉]

[1]Department of Pharmacology, Faculty of Medicine, Kagawa University, Kagawa 761-0793, Japan. [2]Department of Nephrology, Fujita Health University School of Medicine, Aichi 470-1192, Japan. [3]Division of Biomedical Polymer Science, Institute for Comprehensive Medical Science, Fujita Health University School of Medicine, Aichi 470-1192, Japan. [4]Department of Medical Technology, School of Nursing and Medical Care, Yokkaichi Nursing and Medical Care University, Mie 512-8045, Japan. [5]Department of Integrative Genomics, Tohoku Medical Megabank Organization, Tohoku University, Miyagi 980-8573, Japan. [6]Division of Molecular and Cellular Oncology, Miyagi Cancer Center Research Institute, Miyagi 981-1293, Japan. [7]Department of Gastroenterology and Neurology, Faculty of Medicine, Kagawa University, Kagawa 761-0793, Japan. [8]Fujimori Clinic for Internal Medicine and Gastroenterology, Kagawa 761-8075, Japan. [9]Health Science Research Center, Faculty of Medicine, Kagawa University, Kagawa 761-0793, Japan. [10]Department of Innovative Medicine, Graduate School of Medicine, Osaka Medical College, Osaka 569-8686, Japan. [11]Department of Investigative Pathology, Tohoku University Graduate School of Medicine, Miyagi 980-8575, Japan. [12]Department of Applied Life Science, Faculty of Applied Biological Sciences, Gifu University, Gifu 501-1193, Japan. [13]Department of Medical Biophysics, Kobe University Graduate School of Health Sciences, Hyogo 654-0142, Japan. [14]Department of Human Genetics, The University of Tokyo, Graduate School of Medicine, Tokyo 113-0033, Japan. [15]Community Medical Support, Tohoku Medical Megabank Organization, Tohoku University, Miyagi 980-8573, Japan. [16]Department of Cancer Genome Informatics, Faculty of Medicine, Graduate School of Medicine, Osaka University, Osaka 565-0871, Japan. ✉email: afujimoto@m.u-tokyo.ac.jp; akira@med.kagawa-u.ac.jp

