## [Peer Review File · Communications Biology]

Reviewers' comments:

Reviewer #1 (Remarks to the Author):

The paper by Shibyama et al on the possible role of ProRenin Receptor (PRR) in the development of human pancreatic ductal adeno carcinoma (PDAC) is potentially of great interest. This cancer rates about 5th/6th in terms of incidence in different countries worldwide, but current treatment regimes – which include chemotherapy, radiotherapy and surgery – provide only brief remission. One contributor to the poor outcome is the advanced nature of the disease at diagnosis, since the early stages of the disease are asymptomatic. There are some familial forms of the disease, but it is mostly sporadic.

So a paper such as this, which aims to identify key steps in the early progression of PDAC, is very exciting, and particularly relevant perhaps, to the familial form of the disease, where early chemopreventive intervention could be cost effective. Shibyama et al muster an impressive array of advanced techniques to investigate the proposed role of PRR in the development of PDAC. These include forced overexpression of PRR in normal pancreatic cells, in vivo tumorigenicity testing; RNA seq analysis following transfection; DNA analysis for various types of chromosomal damage and rearrangement; chromosomal remodelling, interaction with SMARCA5 – and indeed much more. And, insofar as I can see, all of these experiments were conducted and analysed expertly. The paper is well-written and the amount of data presented is really substantial.

So, it seems mean-minded to be critical. However, the final sentence of para 1 in the Discussion, the authors make a very substantial claim:

“ These data reveal that inappropriate augmentation of (P)RR expression is the fundamental molecular mechanism responsible for the evolution of PDAC”

This is a claim of some importance and if published would herald a whole new direction in pancreatic cancer research and search for therapies. I am genuinely sad to say that I do not think that the evidence presented is sufficient to substantiate this claim. Perhaps the authors have such data but haven't tabled it in the interests of presenting a focused paper - if they have the data, so much the better.

My concerns go back to the start of the paper, to the basic biology and reproducibility of the system.

We are not given much information on the HPDE (which I'm guessing stands for Human Pancreatic Ductal Epithelial) cells – how were they derived? For how many passages or population doublings do they survive normally? Are they readily immortalised and rendered tumorigenic by other plasmids or carcinogens?

The data presented seem to be using mixed cell populations from transfected vs mock transfected; were clones isolated and characterised? How many times were the transfections repeated? If they were repeated were any of the key follow up experiments done on them as well (I appreciate that everything can't be done on every clone)

I appreciate how wonderfully exciting and promising and significant the data looks;
But are we looking at a n=1 experiment??

Are there similar effects if you use totally different expression cassettes?

To ensure specificity of effects, could siRNA or other knockdown experiments not be designed?

Certainly siRNA or antisense on the cell lines could be examined, while accepting that this is not simple as many of the events in carcinogenesis have already happened

I suppose I am saying that to provide a sound basis for such a wide-ranging conclusion as this one, it is necessary to build confidence from a variety of approaches, to ensure adequate repeats, and to give a lot of attention to the basic design of the cell biology experiments; this paper is like a fantastic and impressive stone castle but built on very shaky wooden foundations!

I would encourage the authors to come back with more data – I would love to be proved wrong and for such an exciting and important conclusion to be proved right. But talk alone won't convince me – I need to see repeats and more data

In summary, this is a very interesting paper but I could not recommend its publication unless there is major revision with substantial additional data

Reviewer #2 (Remarks to the Author):

Shibayama et. al. address the role of (pro)renin receptor (PRR) in the development of PDAC. Based on WGS data, the authors conclude that overexpression of PRR in HPDE cells leads to global genomic instability characterized by an increase in SVs, CNAs and SNVs relative to control. They further conclude that this genomic instability is a result of a direct interaction with SMARCA5. The study asks important novel questions that are of interest to the field. However, the following points need to be addressed.

1. The timepoint/passage at which the analysis in figure 1 was conducted is not defined.

2. Gene alterations are known to accumulate with an increasing number of cell divisions. (P)RR expressing cells have 1.8-fold higher cell proliferation as compared to mock. To ensure that the observed genomic changes are a result of (P)RR overexpression, a control from mock cells that have undergone an equivalent number of cell divisions should be included.

3. Fig. 1b: "Compared with Mock cells, cell morphology became more diverse.." Could the authors substantiate this statement further? Eg. did the cells differ in size, shape, N/C ratio, nuclear features?

4. Is the proliferation data in figure 1c significant?

5. The comparisons in figure 3 should be analyzed using hypothesis testing. How does the mock control compare at passage 12?

6. The nuclear atypia in figure 4d,e would benefit from quantification. Was this phenomenon present in the majority of cells?

7. The network analysis suggests an association between SMARCA5 being the mediator of (P)RR effects on expression of various members of the DNA damage response pathway and hence genomic instability. Neither this analysis nor the co-IP experiments prove this association. Would the effects of (P)RR over expression on these proteins persist in cells with siRNA mediated silencing of SMARCA5? The mediator could still be other proteins from the LC/MS-MS analysis, direct (P)RR activity or indirect activity via other known (P)RR functions eg. Wnt/MAPK signaling.

8. Suggestions for figures: Figure 3c does not provide additional data.

The network in supplementary figure 6 is difficult to understand. Perhaps, a table with the contained genes within each subnetwork and a schematic of overlapping nodes would be easier for readers to comprehend.

9. Introduction would benefit from details about (P)RR function in relation to MAPK, Wnt signaling and its various protein domains analyzed in the results section.

Response to comments by reviewers

We thank the reviewers for their thoughtful review and helpful comments, which have guided the revision of the enclosed manuscript. In the response to the variable comments, we have made 39 changes in the revised manuscript as indicated below. We are happy that the manuscript has been improved by this extensive review. We would like to reply, point-by-point, as follows, and the changes made during revision are listed in the last part.

To the comments by reviewer #1

General comments:

So, it seems mean-minded to be critical. However, the final sentence of para 1 in the Discussion, the authors make a very substantial claim:

“ These data reveal that inappropriate augmentation of (P)RR expression is the fundamental molecular mechanism responsible for the evolution of PDAC”, This is a claim of some importance and if published would herald a whole new direction in pancreatic cancer research and search for therapies. I am genuinely sad to say that I do not think that the evidence presented is sufficient to substantiate this claim. Perhaps the authors have such data but haven't tabled it in the interests of presenting a focused paper - if they have the data, so much the better.

Reply

We agreed with the reviewer's comment that our obtained data do not show direct evidence responsible for genetic evolution of PDAC. As we were not able to detect somatic mutations of driver genes as KRAS, TP53, CDKN2A and SMAD4 in (P)RR-expressing HPDE-1/E6E7 cells at the 6th passage, we need to perform a detailed exome analysis in several (P)RR cells at different passages to determine whether driver genes of human PDAC tissue are present or not. We believe that this issue develops into our future research, therefore, we recognize that above description is exaggerated. To avoid confusion, we have changed the last sentence as follows. “These data indicate that inappropriate augmentation of (P)RR expression contributes to the evolution of PDAC.”

In the revised manuscript, we have corrected to “Inappropriate augmentation of (P)RR expression contributes to the evolution of PDAC.” (Please see Changes #6, 12 and 17 in the list of changes below.).

Specific comment #1:

We are not given much information on the HPDE (which I'm guessing stands for Human

Pancreatic Ductal Epithelial) cells – how were they derived? For how many passages or population doublings do they survive normally? Are they readily immortalised and rendered tumorigenic by other plasmids or carcinogens?

Reply

Thank you for your valuable comment. Our HPDE-1 cells were transfected by the E6/E7 gene of human papilloma virus for the purpose of long-term culture and immortalization. HPDE-1/E6E7 cells did not form colonies in soft agar and their cell population was non tumorigenic in nude mice for up to 6 months (Am. J. Path., 1996). We also confirmed that HPDE-1/E6E7 cell population with Mock was not tumorigenic, as shown in Fig. 3. Based on our cell culture system, HPDE-1/E6E7 cells have been propagated up to the 30th passages (unpublished data). Previous studies have shown that HPDE-6 cells were immortalized by the E6/E7 gene of human papilloma virus. It has also been demonstrated that HPDE-6/E6E7 cells are anchorage-dependent and nontumorigenic in SCID mice (Am. J. Path., 2000). In the revised manuscript, we have changed “HPDE” cells to “HPDE-1/E6E7” cells and “HPDE-6/E6E7” cells, respectively for the correct notation.

In the revised manuscript, we corrected HPDE cells to HPDE-1/E6E7 cells and HPDE-6/E6E7 cells (Changes #2, 18 and 27).

Specific comment #2:

The data presented seem to be using mixed cell populations from transfected vs mock transfected; were clones isolated and characterised?

Reply

Thank you very much for indicating an important point. In the present study, we did not isolate clones from each population. We initially compared the level of genomic instability between cell population with Mock and (P)RR to investigate whether aberrant (P)RR expression induces genomic instability or not. As shown in Fig. 1 and Supplementary Fig. 1, Papanicolaou stain shows that Mock cell population is mostly composed of cells with uniformed nucleus and cells with variable nuclear area and multinuclei dominate in (P)RR cell population. Thus, these data indicated that (P)RR expression has a large effect of atypical nuclei on the population level. We recognize that a single cell analysis using (P)RR cell population will be useful to indicate the detailed status of intra-tumoral heterogeneity. Although detail analyses of each clone are beyond the scope of the present study, we do understand that these issues are particularly important. Therefore, we have mentioned that future studies with a single cell analysis would be needed to

support our hypothesis in the Discussion.

In the revised manuscript, we added the data regarding the evaluation of atypical nuclei by Papanicolaou stain, as shown in Fig. 1 and Supplementary Fig. 1 (Changes #2, 27, 29, 37 and 39). We have added descriptions in Abstracts, Results, Discussion and Methods (Changes #3, 7, 13 and 21). According to your suggestion, we also added the following sentence in the Discussion (Change #13). “Diverse atypical nuclei observed in (P)RR-expressing cell populations implies that future studies with a genomic analysis at a single cell level will explain the detailed status of intra-tumoral heterogeneity”. Because we have no experimental data regarding “intra-tumoral heterogeneity”, we have changed to the following description in the 1st paragraph of the Introduction. “In particular, activating mutations of KRAS are almost ubiquitous and inactivating mutations of TP53, SMAD4, CDKN2A, genes related to chromatin modification are also prevalent among PDAC patients²⁻⁴. Recent whole-genome analyses have detected large number of structural variations in PDAC^{4,5}. However, molecular mechanism responsible for genome instability remains unclear (Change #4).”

Specific comment #3:

How many times were the transfections repeated? If they were repeated were any of the key follow up experiments done on them as well (I appreciate that everything can't be done on every clone).

Reply

Thank you for your comment. We prepared the construct of each vector, as shown in Fig.1 and 2. One of the vectors contains the EBNA gene, which enables this vector to transfer to the daughter cell in every cell division. Using this vector, we can maintain cells with stable (P)RR expression at every passage as long as transfected cells are cultured with G418. Namely, the frequency of transfection is just one. The other vector is available for stable and non-replicative transient transfection. Conversely, this vector is not transferred to daughter cells. Transfected cells were cultured with G418 and were directly used for the WGS analysis after one passage. The frequency of transfection is also one in this case. This information is also related to your question #5. We described a brief explanation for each vector in the Figure Legend.

We have added the following sentences to the revised manuscript to explain the character of each vector. “Vector constructs for a stable ATP6ap2 encoding (pro)renin receptor [(P)RR]. EBNA1 gene enables this vector to transfer to daughter cells in every cell division.” in the legend of Fig. 1a (Change #29) and “Vector constructs for stable and non-replicative transient ATP6ap2 encoding (pro)renin receptor. Transfected cells were cultured with G418 for

21 days and analyzed after one passage.” in the legend of Fig. 2a (Change #30). Simple illustrations showing the pattern of each transfection were depicted in Fig. 1a and Fig. 2a (Change #37). We have additionally described Methods for vector constructs for stable and non-replicative transient (P)RR expression (Changes #2, 19 and 27).

Specific comment #4:

I appreciate how wonderfully exciting and promising and significant the data looks; But are we looking at a n=1 experiment??

Reply

We appreciate for your valuable comment. As you pointed out, we prepared for (P)RR-expressing HPDE-6/E6E7 cell population and performed WGS analysis. However, the obtained data had no significant difference between cell population with Mock and (P)RR, because the culture period after passing is two weeks, which resulted in a short period to evaluate genomic instability by WGS. For the purpose of obtaining alternative data, we have performed additional experiments as a Papanicolaou stain to evaluate the level of atypical nuclei using cells cultured for one month. Compared to Mock cells, (P)RR expression significantly led to multinucleated nuclei and much larger and variable nuclear area representing a biological character of cancer in HPDE-1/E6E7 and HPDE-6/E6E7 cell population. Chromosome abnormality was also observed in both populations. These data have been shown in Fig. 1. and Supplementary Fig. 1. As shown in Fig. 2, the WGS analysis was consequently used for the verification of cellular atypia in the revised manuscript. The accuracy of the pipeline used for WGS analysis has been confirmed by Dr. Fujimoto who is an expert in the field (Nature Genetics 2012 and 2016).

In the revised manuscript, we have added the data for atypical nuclei in HPDE-6/E6E7 cell population in Fig. 1 and Supplementary Fig. 1(Changes #29, 37 and 39). For this data, we added the description to Results and Discussion (Changes #7 and 13).

Specific comment #5:

Are there similar effects if you use totally different expression cassettes?

Reply

Thank you for your comment. We have performed WGS using HPDE-1/E6E7 cell population harbouring the vector with stable and non-replicative transient (P)RR expression. By using this

vector, we are able to determine whether different (P)RR expression affects the level of genomic instability. We expected that transient (P)RR expression induces lower genomic instability. Namely, the difference of genomic instability between Mock and (P)RR may become smaller compared to stable (P)RR expression. As shown in Fig. 2, transient (P)RR expression slightly increased the number of somatic mutations and had no effects on SVs. To detect SVs, transient (P)RR expression seems likely insufficient. Compared to transient (P)RR expression, stable (P)RR expression remarkably increased not only the rate of somatic mutation, but also the number of SVs in HPDE-1/E6E7 cell population. Collectively, these data indicate that aberrant (P)RR expression induces a much larger genomic instability as SVs, and support the hypothesis that different (P)RR expression affects the level of genomic instability.

In the revised manuscript, we have added the constructs for stable and non-replicative transient (P)RR expression, as shown in Fig. 2a (Changes #30 and 37). Data for SVs and somatic mutations in HPDE-1/E6E7 cell population with transient (P)RR expression were added in Fig.2b (Change #8). Using the pipeline developed by Dr. Fujimoto (Nature Genetics 2012 and 2016), dry analysis of WGS was performed again in HPDE-1/E6E7 cell population. We additionally indicated driver genes defined by COSMIC detected in transient Mock- and (P)RR-expressing and stable Mock- and (P)RR-expressing HPDE-1/E6E7 cell population. These data are shown in Fig. 2, Supplementary Fig. 2 and Supplementary Table 1, 2 and 3 (Changes #30, 37, 38 and 39). Regarding this data, we completely modified the description in Results, Discussion, Methods and References (Changes #2, 8, 14, 23, 26, 27 and 28). Based on this data, we considered the change of the title in the present study to be appropriate because aberrant (P)RR expression is obviously involved in the development of PDAC (Change #1).

Specific comment #6:

To ensure specificity of effects, could siRNA or other knockdown experiments not be designed? Certainly siRNA or antisense on the cell lines could be examined, while accepting that this is not simple as many of the events in carcinogenesis have already happened.

Reply

Thank you very much for your points. As you pointed out, we have already performed preliminary experiments to investigate whether (P)RR siRNA rescues genomic stability or not. Array CGH was performed in order to examine DNA copy number variation using human PDAC cell line PK-1 treated with scrambled and (P)RR siRNA. Data showed that (P)RR siRNA attenuates genome abnormality on all the chromosome level in PK-1 cells despite of the failure

of DNA repair system. Future studies are underway to elucidate the molecular mechanism regarding this biological phenomenon. Because these data will be also used for patent application, we are unable to describe these preliminary data in the revised manuscript.

Specific comment #7:

I suppose I am saying that to provide a sound basis for such a wide-ranging conclusion as this one, it is necessary to build confidence from a variety of approaches, to ensure adequate repeats, and to give a lot of attention to the basic design of the cell biology experiments; this paper is like a fantastic and impressive stone castle but built on very shaky wooden foundations!

I would encourage the authors to come back with more data – I would love to be proved wrong and for such an exciting and important conclusion to be proved right. But talk alone won't convince me – I need to see repeats and more data. In summary, this is a very interesting paper but I could not recommend its publication unless there is major revision with substantial additional data.

Reply

We thank the reviewer very much for your thoughtful comments. We have made every effort possible to address your concerns. We are very sorry that it took more than 1 year to complete these additional experiments. We believe that substantial additional analyses in response to your suggestions considerably improve the manuscript. Thank you very much again for your thoughtful comments.

To the comments by reviewer #2

Specific comment #1:

The timepoint/passage at which the analysis in figure 1 was conducted in not defined.

Reply

We apologize for our poor explanation. HPDE-1/E6E7 and HPDE-6/E6E7 cells at six-passage were used for Papanicolaou staining and whole genome analysis, respectively (Fig. 1). We have revised the text as follows.

In the revised manuscript, we have added the information of “six- passage” to the data shown in Fig. 1a (Changes #7 and 37) and the legend of Fig. 1a (Change #29).

Specific comment #2:

Gene alterations are known to accumulate with an increasing number of cell divisions. (P)RR expressing cells have 1.8-fold higher cell proliferation as compared to mock. To ensure that the observed genomic changes are a result of (P)RR overexpression, a control from mock cells that have undergone an equivalent number of cell divisions should be included.

Reply

We also understand that genetic alterations are known to accumulate with an increasing number of cell division. However, our data have exhibited that aberrant (P)RR expression substantially induces genomic instability independent of cell division, as shown in Fig. 7. As downregulated molecules induced by (P)RR overexpression are related to cancer (Fig. 5a), we focused on downregulated molecules rather than upregulated ones (Supplementary Fig. 4c). As shown in Fig. 5f and Supplementary Fig. 5c, the expressions of MCM3, PCNA, PARP1 and Ku80, all of which are involved in the maintenance of genomic stability, were downregulated in cells with full-length (P)RR and (P)RR with Δ NTF (N-terminal fragment). The data indicated that DNA replication stress (Fig. 5i and Supplementary Fig. 6) and defects of DNA repair capacity (Fig. 5j and Supplementary Fig. 7) responsible for genomic instability are the most prominent. Thus, the data indicate that aberrant CTF (C-terminal fragment) of (P)RR expression contributes to genomic instability. On the other hand, NTF of (P)RR plays a role in cell proliferation mediated by the activation of Wnt signalling pathway, as shown in Fig. 5g and 5h. However, compared to the cells with Δ NTF of (P)RR, the level of DNA replication stress and the defects of DNA repair capacity were not so considerable in cells with Δ CTF of (P)RR. The data suggest that the

increase of cell division does not predominantly contribute to genomic instability in (P)RR-expressing cells. The scarcity of indels observed in the mutational landscape of (P)RR-expressing cells is also supportive data (Fig. 2), suggesting that genomic instability does not originate from the increase of cell division. We also compared the expressions of MCM3, PCNA, PARP1 and Ku80 between seven-passage FL(P)RR and 10- and 20- passage Mock cells shown in Supplementary Fig. 5d. The data showed that the expression of these molecules persisted even in 20-passage Mock cells that have undergone considerable cell division, which also suggests that genomic instability is not simply caused by the increase of cell division.

In the revised manuscript, the data regarding expression of the components involved in genomic stability pathways and Wnt signalling, cell proliferative ability and DNA damage using cells with each deletion of (P)RR are included in Fig. 5a, f, g, h, and j and Supplementary Fig. 4c, Fig. 5c, Fig. 6 and Fig. 7 (Changes #34, 37 and 39). The comparison of expressions of MCM3, PCNA, PARP1 and Ku80 between seven-passage FL(P)RR and 10- and 20- passage Mock cells are shown in Supplementary Fig. 5d (Change #39). We have added and modified the descriptions related to these data in Results, Discussion and Methods (Changes #10, 15, 22, 25 and 39).

Specific comment #3:

Fig. 1b: "Compared with Mock cells, cell morphology became more diverse." Could the authors substantiate this statement further? Eg. did the cells differ in size, shape, N/C ratio, nuclear features?

Reply

We thank the reviewer for your valuable comments. According to your suggestion, we have evaluated the cell area by Image J in both Mock-and FL(P)RR-expressing HPDE-1/E6E7 and HPDE-6/E6E7 cell population. As shown in Fig. 1, cell area in Mock cells was almost stable. On the other hand, cell area of FL(P)RR-expressing HPDE-1/E6E7 and HPDE-6/E6E7 cell population significantly became variable and much larger. In addition, multinucleated cells and variable and larger nuclear area representing biological characters of invasive cancer were also significant in (P)RR-expressing HPDE-1/E6E7 and-HPDE-6/E6E7 cell population.

In the revised manuscript, the data for the evaluation of cell and nuclear area, as well as, the rate of multinucleated cell are included in Fig. 1 (Changes #29 and 37). For the data, we added the description to Abstracts, Results, Discussion and Methods (Changes #2, 3, 7, 13, 21 and 27).

Specific comment #4:

Is the proliferation data in figure 1c significant?

Reply

Cell proliferative ability was evaluated in HPDE-1/E6E7 cells with Mock and deletion mutants of (P)RR. Compared to Mock cells, cell proliferative ability was significantly greater in cells with FL(P)RR. The proliferative ability of cells with Δ CTF of (P)RR is similar to that of cells with FL(P)RR.

In the revised manuscript, the data have been shown in Fig.5h (Changes #34 and 37) and added the descriptions in Results, Discussion and Methods (Changes #10, 15 and 22).

Specific comment #5:

The comparisons in figure 3 should be analyzed using hypothesis testing. How does the mock control compare at passage 12?

Reply

Actually, we calculated the difference in the number of SNPs between Mock and (P)RR cell population. Data showed that the difference in the number of SNPs became larger at 12-passage. However, we finally decided to delete these data. The reason is as follows. As the number of data is only one in each cell line and the accuracy of genotype detected in the SNP array was not verified by PCR, it seems that these data are not reliable. Since the aim of this study is to elucidate whether aberrant (P)RR expression induces genomic instability or not, we believe that data showing the change of genetic heterogeneity within a cell population is not required for the present study.

In the revised manuscript, we completely deleted the data regarding the number of SNPs between Mock and (P)RR cell population from the original manuscript (Changes #9, 24 and 31). By deleting the data shown in Fig. 3 of the original manuscript, we subsequently changed the order of Figures (Changes #32, 33, 34, 35 and 36).

Specific comment #6:

The nuclear atypia in figure 4d,e would benefit from quantification. Was this phenomenon present in the majority of cells?

Reply

We appreciate your valuable comments. We are unable to perform the quantitative evaluation of nuclear atypia in Mock- expressing HPDE-1/E6E7 cell population *in vivo*, due to no tumor forming in the kidney of immunodeficient mice shown in Fig. 3. Therefore, we simply showed all the pictures of three different tissues formed by (P)RR-expressing HPDE-1/E6E7 cell population, as shown in Supplementary Fig. 3 (Change #39). In the revised manuscript, we have also quantitatively evaluated the nuclear atypia in Mock- and (P)RR-expressing HPDE-1/E6E7 and HPDE-6/E6E7 cell population by Papanicolaou staining. As shown in Fig. 1 and Supplementary Fig. 1, the data demonstrate that (P)RR overexpression significantly leads to multinuclei and the enlargement of nuclear area in both HPDE-1/E6E7 and HPDE-6/E6E7 cell population.

In the revised manuscript, the data showing diverse atypical nuclei in three different tissues are included in Supplementary Fig. 3. Data regarding the quantitative evaluation of atypical nuclei are shown in Fig. 1 and Supplementary Fig. 1 (Changes #29, 37 and 39). We have added the descriptions in Abstracts, Results, Discussion and Methods (Changes #3, 7, 13 and 21).

Specific comment #7:

The network analysis suggests an association between SMARCA5 being the mediator of (P)RR effects on expression of various members of the DNA damage response pathway and hence genomic instability. Neither this analysis nor the co-IP experiments prove this association. Would the effects of (P)RR over expression on these proteins persist in cells with siRNA mediated silencing of SMARCA5? The mediator could still be other proteins from the LC/MS-MS analysis, direct (P)RR activity or indirect activity via other known (P)RR functions eg. Wnt/MAPK signaling.

Reply

In accordance with your suggestion, we have performed the following additional experiments. We examined the effects of SMARCA5 siRNA on (P)RR-overexpressing cells. Data showed that the treatments of SMARCA5 siRNA in both (P)RR-overexpressing HPDE-1/E6E7 and HEK293 cells significantly rescued the expression of molecules such as MCM3, PCNA, PARP1 and Ku80. These data indicate a potential direct molecular interaction between (P)RR and SMARCA5

As we explained in your question #2, our data have indicated that genomic instability is induced by CTF of (P)RR, but not by NTF of (P)RR that regulates Wnt/MAPK signalling pathways

(References# 8, 9, 10 and 11). Thus, it seems likely that genomic instability is not mediated through Wnt/MAPK signalling pathway.

In the revised manuscript, we have added the data regarding the effects of SMARCA5 siRNA on (P)RR-expressing HPDE-1/E6E7 cells to Fig. 6f (Changes #35 and 37) and HEK293 cells to Supplementary Fig. 10d (Change #39). We have added descriptions in the Results, Discussion and Methods (Changes #11, 16 and 20) for these data.

Specific comment #8:

Suggestions for figures: Figure 3c does not provide additional data.

Reply

As explained by the response to your question #5, we have deleted the illustration from Fig. 3c (Changes #9, 24 and 31).

Specific comment #9:

The network in supplementary figure 6 is difficult to understand. Perhaps, a table with the contained genes within each subnetwork and a schematic of overlapping nodes would be easier for readers to comprehend.

Reply

As you have pointed out, the combined molecular network data seem too complicated. Therefore, we have shown each molecular network with high IPA score in Supplementary Fig. 9. Several factors connected to each molecular network exists, as shown in Supplementary Table 4. The data indicate that each molecular network is dependent.

In the revised manuscript, we completely deleted the combined molecular network from the original data. Alternatively, we have added three independent molecular networks composed of the combined molecular networks to Supplementary Fig. 9 (Change #39).

10. Introduction would benefit from details about (P)RR function in relation to MAPK, Wnt signaling and its various protein domains analyzed in the results section.

Reply

As suggested by the reviewer, we have added the detailed descriptions of (P)RR function related to MAPK and Wnt signaling as follows. “(P)RR plays a role of multiple cellular functions. The

specific binding of prorenin and renin to the extracellular domain of (P)RR cleaves angiotensin I from angiotensinogen thus activating renin-angiotensin system⁸. It has also been shown that intracellular signals such as mitogen-activated protein (MAP) kinase are activated by ligands, independent of renin-angiotensin system⁸⁻¹⁰. Moreover, the extracellular domain of (P)RR also has a molecular interaction with low-density lipoprotein receptor protein 6 (LRP6) and Frizzled 8 of Wnt receptor complex¹¹. The activation of Wnt signalling pathway through this molecular interaction is related to the development of PDAC⁷, Glioma¹² and colorectal cancer¹³. However, whether these cellular functions dominantly affect the genomic instability under aberrant (P)RR expression remains to be solved.”

In the revised manuscript, we added the detailed descriptions of (P)RR function related to MAPK and Wnt signalling to the 3rd paragraph of Introduction (Change #5).

List of Changes

Change #1

The title has been changed to clarify the correct meaning of the present study and deleted few words due to words limitation.

Original, Title page:

Aberrant (pro)renin receptor expression induces genomic instability by chromatin remodeller SMARCA5/SNF2H disruption during the progression of pancreatic ductal adenocarcinoma

Revised, Title page:

Aberrant (pro)renin receptor expression induces genomic instability by SMARCA5 disruption during the development of pancreatic ductal adenocarcinoma

Change #2

Additional experiments were performed by Dr. Toru Furukawa for HPDE-6/E6E7 cell studies, Dr. Tsutomu Nakagawa for the vector with transient (P)RR expression and Hiroyuki Ohsaki for Papanicolaou stain. Dr. Jing Hao Wong newly participated in writing this paper. Therefore, their name has been added as co-authors in the revised manuscript. Dr. Fujimoto played a critical role of WGS and added as a co- corresponding author.

Original, Title page, Author's full names:

Yuki Shibayama¹, Kazuo Takahashi², Hisateru Yamaguchi³, Jun Yasuda⁴, Daisuke Yamazaki¹, Asadur Rahman⁵, Takayuki Fujimori⁶, Yoshihide Fujisawa⁷, Shinji Takai⁸, Akihiro Fujimoto⁹, Hideki Kobara⁶, Tsutomu Masaki⁶, Yukio Yuzawa², Hideyasu Kiyomoto¹⁰, Shinichi Yachida¹¹, Akira Nishiyama^{1*}

Revised, Title page, Author's full names and Institutions:

Yuki Shibayama¹, Kazuo Takahashi², Hisateru Yamaguchi^{3,4}, Jun Yasuda^{5,6}, Daisuke Yamazaki¹, Asadur Rahman¹, Takayuki Fujimori^{7,8}, Yoshihide Fujisawa⁹, Shinji Takai¹⁰, Toru Furukawa¹¹, Tsutomu Nakagawa¹², Hiroyuki Ohsaki¹³, Hideki Kobara⁷, Jing Hao Wong¹⁴, Tsutomu Masaki⁷, Yukio Yuzawa², Hideyasu Kiyomoto¹⁵, Shinichi Yachida¹⁶, Akihiro Fujimoto^{14*}, Akira Nishiyama^{1*}

Additional Institutions:

4. Department of Medical Technology, School of Nursing and Medical Care, Yokkaichi Nursing and Medical Care University, Mie, 512-8045, Japan
6. Division of Molecular and Cellular Oncology, Miyagi Cancer Center Research Institute, Miyagi, 981-1293, Japan
8. Fujimori Clinic for Internal Medicine and Gastroenterology, Kagawa, 761-8075, Japan
11. Department of Investigative Pathology, Tohoku University Graduate School of Medicine,

Miyagi, 980-8575, Japan

12. Department of Applied Life Science, Faculty of Applied Biological Sciences, Gifu University, Gifu, 501-1193, Japan

13. Department of Medical Biophysics, Kobe University Graduate School of Health Sciences, Hyogo, 654-0142, Japan

14. Department of Human Genetics, The University of Tokyo, Graduate School of Medicine, Tokyo, 113-0033, Japan

Change #3

In accordance with reviewer's suggestions, abstract has been modified with additional data.

Original, page 3, Abstract:

Aberrant (pro)renin receptor [(P)RR] expression is prevalent in pancreatic ductal adenocarcinoma (PDAC). Here, we investigated whether aberrant expression of (P)RR directly leads to genomic instability in human pancreatic ductal epithelial cells. Whole-genome analysis revealed that aberrant (P)RR expression induced massive chromosomal rearrangements and enhanced the total number of somatic mutations at the global level. (P)RR-expressing cell population exhibited tumour-forming ability, showing both atypical nuclei characterised by distinctive nuclear bodies and the chromosomal abnormalities. (P)RR overexpression upregulated SWI/SNF-related, matrix-associated, actin-dependent regulator of chromatin, subfamily a, member 5 (SMARCA5) through a direct molecular interaction, which resulted in the failure of genomic stability pathways. These data reveal for the first time that aberrant (P)RR expression is marked as an evolutionary origin of PDAC.

Revised, page 4, Abstract:

Aberrant (pro)renin receptor [(P)RR] expression is prevalent in pancreatic ductal adenocarcinoma (PDAC). Here, we investigated whether aberrant expression of (P)RR directly leads to genomic instability in human pancreatic ductal epithelial (HPDE) cells, which are capable of transforming to PDAC. (P)RR-expressing HPDE-1/E6E7 and HPDE-6/E6E7 cells showed obvious cellular atypia harbouring atypical nuclei. Whole genome sequencing revealed that aberrant (P)RR expression induced large numbers of point mutations and structural variations at the genome level. (P)RR-expressing cell population exhibited tumour-forming ability, showing both atypical nuclei characterised by distinctive nuclear bodies and the chromosomal abnormalities. (P)RR overexpression upregulated SWItch/Sucrose Non-Fermentable (SWI/SNF)-related, matrix-associated, actin-dependent regulator of chromatin, subfamily a, member 5 (SMARCA5) through a direct molecular interaction, which resulted in the failure of several genomic stability pathways. These data reveal for the first time that aberrant (P)RR expression contributes the early carcinogenesis of PDAC.

Change #4

To avoid any confusion, descriptions regarding genetic heterogeneity of PDAC have been removed from the original manuscript. Instead, we have described the current status of genomic instability of PDAC.

Original, pages 3-4, First paragraph of Introduction:

Pancreatic ductal adenocarcinoma (PDAC) has a high rate of malignancy, with median survival of 6 months and 5-year survival remaining less than 10%¹. Somatically acquired mutations become a driving force to promote the progression of PDAC. On average, a PDAC patient acquires 67 nonsynonymous mutations². In particular, activating mutations of KRAS are almost ubiquitous and inactivating mutations of TP53, SMAD4 and CDKN2A are also prevalent among PDAC patients²⁻⁴. In addition, recent whole-genome analyses in PDAC patients have detected mutations in genes related to chromatin modification, DNA damage repair and molecular mechanisms promoting carcinogenesis. However, no particularly common gene has been identified in PDAC patients^{3,4}. Among the genes shown to be associated with PDAC, there is a large number of infrequently mutated genes, reflecting significant genetic heterogeneity⁵. Although this genetic heterogeneity confers resistance to therapy on PDAC, its fundamental molecular mechanism remains unclear⁶.

Revised, pages 4-5, First paragraph of Introduction:

Pancreatic ductal adenocarcinoma (PDAC) has a high rate of malignancy, with median survival of 6 months and 5-year survival remaining less than 10%¹. Somatically acquired mutations become a driving force to promote the progression of PDAC. On average, a PDAC patient acquires 67 nonsynonymous mutations². In particular, activating mutations of KRAS are almost ubiquitous and inactivating mutations of TP53, SMAD4, CDKN2A, genes related to chromatin modification are also prevalent among PDAC patients²⁻⁴. Recent whole-genome analyses have detected large numbers of structural variations in PDAC^{4,5}. However, molecular mechanism responsible for genome instability remains unclear.

Change #5

We have added new paragraph to explain the cellular function of (P)RR related to MAPK and Wnt signalling.

Revised, pages 5-6, Third paragraph of Introduction:

(P)RR plays a role of multiple cellular functions. The specific binding of prorenin and renin to the extracellular domain of (P)RR cleaves angiotensin I from angiotensinogen thus activating renin-angiotensin system⁸. It has also been shown that intracellular signals such as

mitogen-activated protein (MAP) kinase are activated by ligands, independent of renin-angiotensin system⁸⁻¹⁰. Moreover, the extracellular domain of (P)RR also has a molecular interaction with low-density lipoprotein receptor protein 6 (LRP6) and Frizzled 8 of Wnt receptor complex¹¹. The activation of Wnt signalling pathway through this molecular interaction is related to the development of PDAC⁷, Glioma¹² and colorectal cancer¹³. However, whether these cellular functions dominantly affect the genomic instability under aberrant (P)RR expression remains to be solved.

Change #6

To describe more precisely, we have modified the sentence, as described below.

Original, page 5, Last sentence of Introduction:

These data reveal that inappropriate augmentation of (P)RR expression is marked as an evolutionary origin of PDAC.

Revised, page 6, Last sentence of Introduction:

The data reveal that inappropriate augmentation of (P)RR expression contributes to the genetic evolution of PDAC.

Change #7

We have added new results for cellular atypia in HPDE-1/E6E7 and HPDE-6/E6E7 cell population overexpressed by either Mock or (P)RR cell population We have completely changed the description for Fig. 1.

Original, pages 5-7, Results, Massive chromosomal rearrangements by (P)RR overexpression:

We established (P)RR-overexpressing HPDE cells using a vector inserted with the coding sequence of ATP6ap2 encoding (P)RR to perform the following experiments. The insertion of ATP6ap2 was confirmed by PCR (Fig. 1a). As a control, we also included cells transfected by a vector without the insertion of ATP6ap2, referred to as Mock (Fig. 1a). Compared with Mock cells, cell morphology became more diverse owing to aberrant (P)RR expression (Fig. 1b). Cell proliferative ability was increased by (P)RR overexpression (Fig. 1c). To determine the level of chromosomal rearrangements at the whole-genome level, we analysed the structural variation (SV), such as inter- and intratranslocation, inversion, deletion, and insertion, using BreakDancer ver. 1.110 by human whole-genome sequencing. To display the distribution of SVs in the whole genome, chromosomes were split into windows of specific size. Although distinctive SVs were present below 1,000 bp as a window size in Mock, for window

sizes above this level, SVs were detected only in (P)RR, with negligible levels in Mock (Fig. 1d and Supplementary Fig. 1). However, at the window size of 1,000 bp, deletion (DEL: 870 in total; Fig. 1e, f) was the most common chromosomal rearrangement in cells with high (P)RR expression, suggesting substantial breakpoints. Intrachromosomal rearrangement (CTX: 257 in total), interchromosomal rearrangement (ITX: 705 in total), inversion (INV: 158 in total) and insertion (INS: 25 in total) were also distributed throughout the chromosomes (Fig. 1e, f). In the case of over 1,000 bp as a window size, the total number of chromosomal rearrangements was 1,702 in (P)RR and 57 in Mock at 10,000 bp (Supplementary Fig. 1c) and 1,378 in (P)RR and 42 in Mock at 100,000 bp (Supplementary Fig. 1d). At a window size of 1,000 bp, the number of SVs per 100 Mbp in each chromosome was substantially greater in (P)RR cells than in Mock cells (Fig. 1g). By using cells with Mock as a reference genome, DNA copy number variation (CNV) was also identified throughout the lengths of almost all of the chromosomes in cells with (P)RR. However, CNVs commonly detected in PDAC⁴ (e.g., *CDKN2A*, *SMAD4*, *KDM6A*, *SMARCA2* and *RPA1*) were not detected in six-passage cells with (P)RR (Supplementary Table 1). Overall, these data demonstrate that aberrant (P)RR expression leads to massive chromosomal rearrangements, such as translocation and inversion, at the global genome level irrespective of window size in HPDE cells.

Revised, pages 6-7, Results, Aberrant (P)RR expression generates cellular atypia in HPDE-1/E6E7 and HPDE-6/E6E7 cell population:

We established (P)RR-overexpressing HPDE-1/E6E7¹⁵ and HPDE-6/E6E7¹⁶ cells using a stable vector inserted with the coding sequence of *ATP6ap2* encoding (P)RR to perform the following experiments. As a control, we included cells transfected by a vector without the insertion of *ATP6ap2*, referred to as Mock (Fig. 1a). The insertion of *ATP6ap2* was confirmed in both cells at six-passage by PCR (Fig. 1a). Compared to cells with Mock, (P)RR overexpression significantly induced much larger and variable cell area ($P < 0.0001$ vs. Mock) in HPDE-1/E6E7 and HPDE-6/E6E7 cell population. (P)RR overexpression also exhibited nuclei with distinctive nuclear bodies (Fig. 1b). To evaluate the level of atypical nuclei in HPDE-1/E6E7 and HPDE-6/E6E7 cells, we performed Papanicolaou stain. In both cell populations under aberrant (P)RR expression, multinucleated cells were considerable and the increase of (P)RR expression significantly led to larger and variable nuclear area ($P < 0.0001$ vs. Mock; Fig. 1c and Supplementary Fig. 1). Chromosome abnormality was also observed in both cell populations expressing (P)RR (Fig. 1c and Supplementary Fig. 1). It is clear from the data that (P)RR-expressing HPDE-1/E6E7 and HPDE-6/E6E7 cell population generates cells harbouring diverse atypical nuclei.

Change #8

We have added the data for the comparison of genomic instability between transient and stable (P)RR-expressing HPDE-1/E6E7 cell population analysed by WGS. We have changed all the description regarding Fig. 2.

Original, pages 7-8, Results, Mutational landscape in (P)RR-overexpressing cell population:

Based on the variant call calculated using Bcftools11 under human whole-genome analysis (Supplementary Table 2), we also determined the level of total somatic mutations in six-passage HPDE cells expressing either Mock or (P)RR. Total somatic mutations detected in cells with (P)RR were approximately twofold higher than in Mock cells [Mock: 124,579; (P)RR: 239,080; single-nucleotide polymorphisms (SNPs) registered in dbSNP build 144 were excluded from the total somatic mutations]. As shown by a Venn diagram, the total numbers of single-nucleotide variations (SNVs), insertions (INSs) and deletions (DELs) were approximately 1.5-, 2.3- and 2.1-fold higher in (P)RR-overexpressing cells, respectively (Fig. 2a). The accuracy of SNVs identified by Bcftools was confirmed to be more than 98% by SNP microarray analysis. Somatic mutations acquired by (P)RR overexpression were distributed throughout all of the chromosomes (Fig. 2b). The number of somatic mutations per 100 Kbp in cells with (P)RR was increased in all of the chromosomes (Fig. 2b). The numbers of somatic mutations of eight significant genes detected in PDAC patients³ [Mock: 16; (P)RR: 39; Fig. 2c ; Supplementary Table 3] and BRCA pathway genes involved in DNA maintenance⁴ [Mock: 11; (P)RR: 48; Fig. 2d ; Supplementary Table 4] were also increased by 2.6- and 4.4-fold, respectively, which also became more diverse in (P)RR cells (Fig. 2c, d). Additionally, the total number of significant mutations detected in the exons of (P)RR cells was 2.9-fold greater than those in Mock cells [Mock: 23; (P)RR: 67; Fig. 2e]. The number of missense mutations in (P)RR cells was also increased by 1.8-fold [Mock: 14; (P)RR: 25; Fig. 2e]. Nonsense and frameshift mutations that have deleterious effects were present only in (P)RR cells but not in Mock cells (Fig. 2e). Finally, somatic mutations within the exons of genes commonly associated with PDAC and BRCA pathway genes were not found in six-passage Mock or (P)RR cells (Supplementary Table 5). These data reveal that an increase in (P)RR expression promotes the accumulation of somatic mutations at the whole-genome level.

Revised, pages 7-9, Results, Genomic instability in HPDE-1/E6E7 cell population with transient and stable (P)RR expression:

To compare genomic instability between HPDE-1/E6E7 cell population with transient and stable (P)RR expression, we also established (P)RR-overexpressing HPDE-1/E6E7 cells using a stable and non-replicative transient vector inserted with the coding sequence of ATP6ap2¹⁷ (Fig. 2a). The insertion of ATP6ap2 was confirmed in HPDE-1/E6E7 cells with transient (P)RR expression by PCR (Fig. 2a).

We performed whole genome sequencing for untreated, transient Mock- and (P)RR-overexpressing and stable Mock- and (P)RR-overexpressing HPDE-1/E6E7 cells. By comparing each Mock and (P)RR-overexpressing cells against untreated HPDE-1/E6E7 cells, we detected point mutations, short insertions and deletions (indels), and structural variations (SVs). Our analyses identified much larger numbers of point mutations and SVs in stable (P)RR-overexpressing cells than other treated cells. Furthermore, stable (P)RR expression against transient (P)RR considerably induced higher numbers of somatic mutations and SVs than stable Mock expression against transient Mock. These data indicate that the level of (P)RR expression affects the difference in genomic instability (Fig. 2b, c, Supplementary Fig. 2 and Supplementary Table 1). Stable (P)RR-overexpressing cells increased the number of point mutations and SVs than stable Mock cells by 6.5- and 8.8- fold, respectively. Chromosomal translocations detected in the stable (P)RR-overexpressing cells numbered 122 and dominated in all the SVs (48%). However, there was no difference in the number of short indels among treated cells.

We next focused on protein alternating mutations in the stable (P)RR-overexpressing cells. Our analyses identified 63 nonsynonymous mutations, and known driver genes defined by the COSMIC database (FGFR3, MLL3, BRIP1 and MSH6) were included (Supplementary Table 2). In the SVs, breakpoints were detected in 5 COSMIC driver genes (PDE4DIP, THRAP3, FANCD2, MDS1 and CBL; Supplementary Table 3). Additionally, a chromosomal translocation was detected in a region close to a LINE1 transposable element located on intron1 of TTC28. Recurrent chromosomal translocation events in this region were reported in various cancers and were considered to be caused by LINE1 transposition¹⁸⁻²⁰. Another chromosomal translocation was observed in MACROD2 gene, which is located at a fragile site and can be related with DNA repair system through PARP1 poly ADP-ribosylation (PARylation)^{21,22}. These results suggest that (P)RR overexpression induces mutations in some important driver genes, which may contribute to genetic abbreviation in cancer.

Change #9

As suggested by the reviewer, we have deleted the data related to genetic heterogeneity in the cell population.

Original, pages 8-9, Results, (P)RR-expressing cells develop a heterogeneous phenotype:

To determine the extent to which genetic diversification progressed by comparing cells overexpressing (P)RR at different passages, we performed an exome SNP microarray in each of Mock and (P)RR cells. We confirmed significant (P)RR overexpression in 12-passage cells by PCR (Fig. 3a, b). With an increasing number of passages for HPDE and HEK293 cells, the total

number of SNPs detected only in (P)RR-expressing cells substantially increased by 4.5- and 2.0-fold, respectively [HPDE; (P)RR P6: 22; (P)RR P12: 99; HEK293; (P)RR P6: 71; (P)RR P12: 144; Fig. 3a, b], as did the number of missense mutations [HPDE; (P)RR P6: 16; (P)RR P12: 86; HEK293; (P)RR P6: 64; (P)RR P12: 134; Fig. 3a, b]. These findings demonstrate that cell populations with (P)RR experience clonal evolution with an increased number of passages (Fig. 3a, b; Supplementary Table 6), as depicted in a schematic summary (Fig. 3c).

Revised:

Deleted.

Change #10

We have modified the descriptions more precisely to elucidate the molecular mechanism responsible for genomic instability induced by (P)RR overexpression.

Original, pages 11-14, Results, Downregulation of genomic stability pathways induced by aberrant (P)RR expression:

To elucidate the molecular mechanism behind the genomic instability induced by aberrant (P)RR expression, we performed LC-MS/MS analysis in the fraction of insoluble nucleus extracted from Mock- and (P)RR-expressing HPDE cells. By this analysis, we detected differences in the expression of a total of 14,583 peptides and significant differences for 2,340 peptides between the two cells by a global permutation-based false discovery rate (FDR) approach implemented in Perseus¹⁵ (Supplementary Fig. 2a, b). Based on the fact that CTF of (P)RR was expressed in the insoluble nucleus containing chromatin binding protein, further analyses were performed by using biological information regarding chromatin and/or DNA binding as Gene Ontology (GO) terms¹⁶ (Supplementary Table 7). Using significant peptides filtered by GO terms, we also performed Ingenuity Pathway Analysis (IPA) to identify each molecular function and canonical pathways affected by the increase of (P)RR expression. By analyses of upregulated and downregulated molecules, it was revealed that (P)RR overexpression substantially affects gene expression and DNA replication, recombination and repair (Fig. 6a and Supplementary Fig. 2c). The significant canonical pathways responsible for Granzyme A signalling and oestrogen receptor signalling were identified by analysis of the upregulated molecules (Supplementary Fig. 2d). Furthermore, different genome stability pathways such as cell cycle control of chromosomal replication, base excision repair (BER), DNA double-strand break repair by nonhomologous end joining (NHEJ) and telomere extension by telomerase (Fig. 6b, c) pathways were identified with high IPA statistical significance. We also confirmed the molecular functions of the downregulated molecules. Besides the inactivation of the components responsible for DNA damage response such as H2AX (Ser.139)

and p53 (Ser.15) (Supplementary Fig. 3a), we also confirmed the significant downregulation of MCM3, PCNA, PARP1 and Ku80 encoded by XRCC5 in both HPDE and HEK293 cells with (P)RR overexpression (Fig. 6d and Supplementary Fig. 3b). By using cells with the deletion of each domain of (P)RR (Fig. 6e), we further conducted DNA fibre assay to measure the progression of individual replication forks, fork stalling and asymmetry between sister replication forks.

Revised, pages 11-13, Results, Downregulation of genomic stability pathways induced by aberrant (P)RR expression:

To elucidate the molecular mechanism responsible for the genomic instability induced by aberrant (P)RR expression, we performed LC-MS/MS analysis in the fraction of insoluble nucleus extracted from Mock- and (P)RR-expressing HPDE-1/E6E7 cells. From this analysis, we detected differences in expression of a total of 14,583 peptides. We implemented the global permutation-based false discovery rate (FDR) approach was implemented in Perseus24 and 2,340 peptides with FDR=0.05 and minimal fold change (S0)=0.5 were considered as significantly differentiated peptides (Supplementary Fig. 4a,b). Based on the fact that CTF of (P)RR was expressed in the insoluble nucleus containing chromatin binding protein, further analyses were performed by using biological information regarding chromatin and/or DNA binding as Gene Ontology (GO) terms²⁷ (Supplementary Table 4). Using significant peptides filtered by GO terms, we also performed Ingenuity Pathway Analysis (IPA) to identify related diseases, and each molecular function and canonical pathways affected by the increase of (P)RR expression. Regarding downregulated molecules, cancer was identified as the top of related diseases and it was not expected from the information of upregulated molecules (Fig. 5a and Supplementary Fig. 4c). By analyses of upregulated and downregulated molecules, it was revealed that (P)RR overexpression substantially affects gene expression and DNA replication, recombination and repair (Fig. 5b and Supplementary Fig. 4d). The significant canonical pathways responsible for “Granzyme A signalling”, “estrogen receptor signalling”, “nucleotide excision repair (NER)” and “glucocorticoid receptor signalling” were identified by analysis of the upregulated molecules (Supplementary Fig. 4e). On the other hand, different genome stability pathways such as “cell cycle control of chromosomal replication”, “base excision repair (BER)”, “DNA double-strand break repair by nonhomologous end joining (NHEJ)” and “telomere extension by telomerase” pathways (Fig. 5c, d) were identified with high statistical significance under IPA in the downregulated molecules. As these pathways are marked as main pathways implicated in genomic instability²⁸, we also examined molecular functions composed of the downregulated molecules by using cells with the deletion of each domain of (P)RR (Fig. 5e and Supplementary Fig. 5a). Besides the inactivation of the components responsible for DNA damage response such as H2AX (Ser.139) and p53 (Ser.15) by (P)RR overexpression

(Supplementary Fig. 5b), we also confirmed significant downregulation of MCM3, PCNA, PARP1 and Ku80 encoded by XRCC5 in both HPDE-1/E6E7 and HEK293 cells with FL(P)RR and (P)RR-ΔN at six-passage (Fig. 5f and Supplementary Fig. 5c). Even in 20-passage Mock cells having undergone a substantial cell division, expression of these molecules was maintained (Supplementary Fig. 5d). On the other hand, Wnt components such as pLRP6 and active β-catenin were upregulated in cells with FL(P)RR and (P)RR-ΔC (Fig. 5g). Cell proliferative ability was also significantly increased in these cells (Fig. 5h). Collectively, these results indicate that downregulation of molecules involved in genomic stability mediated by CTF of (P)RR is not associated with the increased number of cell division through NTF of (P)RR. We also conducted DNA fibre assay to determine the progression of individual replication forks and fork stalling.

Change #11

The results of the effects of SMARCA 5 siRNA on (P)RR-overexpressing cells have been updated.

Original, page 15, Results, Direct molecular binding of (P)RR with SMARCA5 to form an ISWI chromatin remodelling complex

The data showed that endogenous (P)RR undertook direct molecular binding with endogenous SMARCA5 (Fig. 7e and Supplementary Fig. 7c).

Revised, page 16, Results, Direct molecular binding of (P)RR with SMARCA5 to form an ISWI chromatin remodelling complex

The data showed that endogenous (P)RR undertook direct molecular binding with endogenous SMARCA5 (Fig. 6e and Supplementary Fig. 10c). The treatment of SMARCA5 siRNA rescued the expression of molecules responsible for genomic stability pathways in (P)RR-expressing cells (Fig. 6f and Supplementary Fig. 10d).

Change #12

The last sentence has been modified accordingly.

Original, page 16, First paragraph of Discussion:

These data reveal that inappropriate augmentation of (P)RR expression is the fundamental molecular mechanism responsible for the evolution of PDAC.

Revised, page 17, First paragraph of Discussion:

These data reveal that inappropriate augmentation of (P)RR expression is the fundamental molecular mechanism responsible for the early carcinogenesis of (P)RR in PDAC.

Change #13

We have added discussion about cellular atypia in HPDE-1/E6E7 (Fig. 1).

Original, pages, 16-17, Second paragraph of Discussion:

Human whole-genome analyses have revealed that aberrant (P)RR expression is capable of generating substantial SVs representing inter- and intrachromosomal translocation, inversion and CNVs. In contrast to Mock-expressing HPDE cells, the complex chromosomal rearrangements were stably maintained in (P)RR-expressing HPDE cells through different window size. Chromosomal rearrangements such as translocations and inversions are common in PDAC patients^{4,32}. Interestingly, the pattern of chromosomal rearrangements detected in cells with aberrant (P)RR expression was quite similar to the “unstable subtype” observed in PDAC patients, which is characterized by the largest chromosomal rearrangements at the global level¹⁴. Complex genomic rearrangement patterns, such as chromothripsis³³ and polyploidization linked to unstable tumour³⁴, were also confirmed in two-thirds of PDAC patients by using a validated informatic tool named CELLULOID, which estimates tumour ploidy and copy number from whole-genome data³². These analyses explained the punctuated equilibrium rather than the gradualism defined by PanIN progression in the evolutionary processes of PDAC¹⁹. Taking the obtained findings together, the detailed analyses of the complex chromosomal rearrangement in the (P)RR-expressing cell population support the “catastrophic model”³⁵ as a form of punctuated equilibrium revealing the mutational processes required for the progression of PDAC.

Revised, page 17, Second paragraph of Discussion:

Our previous study showed that the increase of (P)RR expression synchronizes with the appearance of atypical nuclei observed in PanIN-2 of human PDAC tissues⁷. In the present study, (P)RR-expressing HPDE-1/E6E7 and HPDE -6/E6E7 cell population was mainly composed of significantly larger cells containing atypical nuclei with distinctive nuclear bodies, much variable and larger nuclear area, multinucleate as well as abnormal chromosomes, all of which are biological characters of cancer. Diverse atypical nuclei observed in (P)RR-expressing cell populations implies that future studies with genomic analysis at a single cell level will explain the detailed status of intra-tumoral heterogeneity.

Change #14

We organized discussion for WGS analysis in the revised manuscript. On the other hand, Discussion for tumor-forming ability in HPDE-1/E6E7 cell population with (P)RR was moved to the fourth paragraph.

Original, pages 17-18, Third paragraph of Discussion:

Our data also demonstrate that aberrant (P)RR expression enhances not only the accumulation of gene alterations including SNV, INS and DEL throughout all of the chromosomes, but also the number of nonsynonymous mutations. The observation of dominant background mutations and massive chromosomal rearrangements in (P)RR-expressing HPDE cells reflect the occurrence of transient mutational bursts³⁴ and the possibility of creating evolutionary opportunities³⁶. In the present study, nonsynonymous mutations in PDAC⁴ and BRCA pathway⁴ genes responsible for genomic stability were not detected in early-passage (P)RR-expressing HPDE cells. However, the total number of gene mutations was significantly increased by aberrant (P)RR expression, suggesting a potential contribution of (P)RR to the development of genetic heterogeneity. Additionally, SNPs in exons were significantly increased during passages in the cell population with (P)RR overexpression. These findings strongly indicate that aberrant (P)RR expression induces and generates a cell population with the potential for malignancy. These biological phenomena support a potential model representing the carcinogenesis of PDAC⁶⁻³⁷. Indeed, the present in vivo analysis also showed that the (P)RR-expressing HPDE cell population exhibited tumour-forming ability. Although the explosive cell population growth was not observed owing to the absence of KRAS codon 12 mutation, (P)RR-expressing cells exhibited atypical nuclei with the chromosomal abnormalities of bridge and fusion generated by telomere dysfunction¹². These findings are consistent with the concept that (P)RR-expressing HPDE cell populations undergo a precancerous state and have the potential to progress towards cancerous development. Because theoretical analyses have shown that gene alterations accumulate in proportion to the number of cell divisions^{6,37} and the level of acquired neutral mutations³⁸, long-term cell culture until the successful detection of distinctive nonsynonymous mutation of PDAC patients may become an effective methodology for cancerous development using a (P)RR-expressing HPDE cell population.

Revised, pages 17-18, Third paragraph of Discussion:

Whole-genome sequencing has revealed that stable (P)RR overexpression induces large numbers of point mutations and SVs including driver genes defined by COSMIC, but does not increase indels. These results indicate that stable (P)RR overexpression induces genomic instability responsible for the generation of tumour heterogeneity. Several studies have shown that increased SVs and somatic mutations are frequently observed in PDAC^{4,5}. Furthermore, SVs associated with the appearance of driver genes of PDAC and the increase of total somatic mutations were found in PanIN^{5,41}. Additionally, PDAC patients with large number of SVs called as “unstable subtype” were characterized by more aggressive behaviour, which is related to refractory^{4,5}. Indeed, (P)RR expression was increased with severe stage in the TNM classification in tissues of PDAC patients⁶. Taken together, these results suggest that sequential

elevation of (P)RR expression contributes not only to carcinogenesis, but also the augmentation of aggressiveness and refractory representing a malignancy of PDAC patients through genetic aberration.

Revised, pages 18-19, Fourth paragraph of Discussion:

In the present study, (P)RR-expressing HPDE-1/E6E7 cell population exhibited tumour-forming ability. Although the explosive cell population growth was not observed owing to the absence of KRAS codon 12 mutation, (P)RR-expressing cells exhibited atypical nuclei with chromosomal abnormalities of bridge and fusion generated by telomere dysfunction²³. These findings are consistent with the concept that (P)RR-expressing HPDE-1/E6E7 cell populations undergo a precancerous state and have the potential to progress towards becoming cancerous. Because theoretical analyses have shown that gene alterations accumulate in proportion to the number of cell divisions^{42,43} and the level of acquired neutral mutations⁴⁴, future studies with long-term cell culture will be needed until the successful detection of driver mutations of PDAC patients in (P)RR-expressing HPDE-1/E6E7 cell population.

Change #15

We have added the discussion for the fact that genomic instability induced by aberrant (P)RR expression is not caused by the increase of cell division. Discussion for NHEJ and telomere length has been moved to the seventh paragraph.

Original, pages 19-20, Fifth paragraph of Discussion:

It has also been shown that DNA replication stress leads to the accumulation of DNA lesions and induces striking chromosomal instability in cancer cells in the absence of mitotic dysfunction⁴⁴. Furthermore, loss of function of BER enhances the somatic mutation rate⁴⁵. In the present study, whole-genome analyses indicate that the above genomic changes are induced by aberrant (P)RR expression.

Revised, page 20, Sixth paragraph of Discussion:

It has also been shown that DNA replication stress leads to the accumulation of DNA lesions and induces striking chromosomal instability in cancer cells in the absence of mitotic dysfunction⁴⁹. Interestingly, our data have revealed that aberrant (P)RR expression induces DNA replication stress and SVs including several dozens of large deletions, suggesting DNA replication stress is not due to mitotic dysfunction. Namely, domain of (P)RR regulating genomic stability including DNA replication is mediated by CTF of (P)RR, which is independent of NTF of (P)RR induced cell proliferation. The scarcity of indels observed in the mutational landscape of stable (P)RR-overexpressing cells also supports the concept that DNA replication stress is not mediated by cell proliferation. Since loss of function of BER enhances

the somatic mutation rate⁵⁰, massive SVs and elevated rate of somatic mutations detected by whole -genome sequencing may be the consequence of the above genetic change induced by (P)RR overexpression.

Revised, pages 20-21, Seventh paragraph of Discussion:

Dysfunction of NHEJ leads to failure of regulation of the correct length of telomeres and of protection of their ends⁵¹. In association with the reduction of Ku80 expression⁵², shortened telomeres are observed as an initial genetic change, which are found in more than 90% of PDAC tissues^{31,53,54}. Consistent with these pathological analyses in PDAC patients, the present study reveals that aberrant (P)RR expression reduces the telomere length associated with the downregulation of XRCC5 encoding Ku80, which is associated with complex SVs. These findings support our hypothesis that an increase of (P)RR expression is an essential molecular mechanism behind the early carcinogenesis in PDAC. In this context, our data previously indicated that plasma-soluble (P)RR is a potential biomarker to identify patients with PDAC from an early stage⁷.

Change #16

We have added the Results and Discussion regarding the effect of SMARCA5 siRNA on (P)RR-expressing cells to the revised manuscript.

Original, page 21, Sixth paragraph of Discussion:

In the present study, we have demonstrated a direct molecular binding between (P)RR and SMARCA5. These data support the hypothesis that the upregulation of SMARCA5 by aberrant (P)RR expression through a direct molecular interaction plays an important role in generating genetic evolution.

Revised, page 21, Eighth paragraph of Discussion:

In the present study, we have demonstrated a direct molecular binding between (P)RR and SMARCA5. Furthermore, reduction in SMARCA5 rescued the expression of molecules responsible for genomic stability in (P)RR-expressing cells. These data support the hypothesis that the upregulation of SMARCA5 is induced by aberrant (P)RR expression through their direct molecular interaction, which plays an important role in generating genetic evolution of PDAC.

Change #17

In the Discussion, the last sentence has been modified to describe the fact more precisely.

Original, page 23, Eighth paragraph of Discussion:

These data reveal that aberrant (P)RR expression is marked as an evolutionary origin of PDAC.

Revised, page 23, Tenth paragraph of Discussion:

These data indicate that inappropriate augmentation of (P)RR expression contributes to the evolution of PDAC.

Change #18

We have added the information of HPDE-6/E6E7 cells and modified from HPDE cell to HPDE-1/E6E7 and HPDE-6/E6E7 cells throughout the revised manuscript.

Original, page 23, Methods, Cell lines:

HPDE cells⁵⁸ were cultured in Hu-Media KG2 (Kurabo, Osaka, Japan; catalogue #KK-2150S).

Revised, page 23, Methods, Cell lines:

Immortalized HPDE-1/E6E7¹⁵ and HPDE-6/E6E7¹⁶ cells were cultured in Hu-Media KG2 (Kurabo, Osaka, Japan; catalogue #KK-2150S).

Change #19

We have added the information of vector with stable and non-replicative transient (P)RR expression in the revised manuscript.

Original, pages 24, Methods, Vector construction:

A DNA fragment corresponding to human ATP6ap2 (1,050 bp without a stop codon) encoding (P)RR was inserted into the restriction sites of BamHI and AgeI within the mammalian expression plasmid pEB Multi-Neo TARGET tag-C (Wako, Osaka, Japan; catalogue #165-26521) to generate both (P)RR-expressing HPDE and HEK293 cells.

Revised, page 24, Methods, Vector construction:

A DNA fragment corresponding to human ATP6ap2 (1,050 bp without a stop codon) encoding (P)RR was inserted into the restriction sites of BamHI and AgeI within the mammalian expression plasmid pEB Multi-Neo TARGET tag-C (Wako, Osaka, Japan; catalogue #165-26521) to generate both stable (P)RR-expressing HPDE-1/E6E7 and HPDE-6/E6E7 and HEK293 cells. Human ATP6ap2 inserted into pcDNA 3.0 tagged with 10His¹⁷ was also used for making stable and non-replicative transient (P)RR-expressing cells.

Change #20

We have added new protocols for the transfection of SMARCA 5 siRNA into (P)RR expressing cells in the revised manuscript. We have also changed the caption from “Plasmid transfection” to “Plasmid and siRNA transfection”.

Original, page 25, Methods, Plasmid transfection:

In accordance with the manufacturer’s instructions, FL(P)RR-, (P)RR- Δ N-, (P)RR- Δ C- or SMARCA5-expressing vector or an empty vector was transfected into HPDE and HEK293 cells using Lipofectamine 3000 (Thermo Fisher Scientific; catalogue #L3000008).

Revised, pages 25, Methods, Plasmid and siRNA transfection:

In accordance with the manufacturer’s instructions, FL(P)RR-, (P)RR- Δ N-, (P)RR- Δ C- or SMARCA5-expressing vector or an empty vector was transfected into HPDE-1/E6E7 and HEK293 cells using Lipofectamine 3000 (Thermo Fisher Scientific; catalogue #L3000008). Using Lipofectamine RNAiMAX (Thermo Fisher Scientific; catalogue #13778075), SMARCA5 Stealth RNAiTM siRNA was transfected into (P)RR-expressing cells according to the recommended protocol. The sequence information in primers of SMARCA5 Stealth RNAiTM siRNA was 5’-GGA GGC UUG UGG AUC AGA AUC UGA A-3’ and 5’-UUC AGA UUC UGA UCC ACA AGC CUC C-3’. For the scrambled siRNA, Stealth RNAiTM siRNA Negative Control Med GC (Thermo Fisher Scientific; catalogue #12935-300) was used.

Change #21

We have added new section for the whole protocol of cell atypia (Fig. 1).

Revised, page 26, Methods, Cellular atypia:

We evaluated the cell area of HPDE-1/E6E7 and HPDE-6/E6E7 cells expressing Mock or (P)RR using Image J. Papanicolaou stain was also performed. Multinucleated cells were evaluated by visual observation and the nuclear area was measured by Image J in Mock and (P)RR-expressing HPDE-1/and- HPDE-6/E6E7 cells.

Change #22

We have modified the sentence for transfected cells evaluated by direct cell counting.

Original, page 26, Methods, Direct cell counting:

We performed a count of the total number of HPDE cells expressing either Mock or (P)RR over 28 days.

Revised, page 27, Methods, Direct cell counting:

We performed a count of the total number of HPDE-1/E6E7 cells expressing either Mock or the deletion of each domain of (P)RR over 18 days.

Change #23

We have added the protocols for wet WGS analyses in HPDE-1/E6E7 cell population with stable and non-replicative transient (P)RR expression. Additionally, we have completely modified the protocols for dry WGS analysis. Wet and dry analyses have been integrated into the same section entitled with “Whole -genome sequencing and mutation calling “in the revised manuscript.

Original, page 34, Methods, Whole -genome sequencing:

In accordance with the cBot User Guide Rev. L (Part #15006165), whole-genome libraries were prepared for cluster generation by cBot. To analyse a sequence on the next-generation sequencer, the flow cells were clustered on the cBot using HiSeq PE Cluster Kit v4 cBot (Illumina ;catalogue #PE-401-4001). DNA libraries were analysed on the Illumina HiSeq 2500 instrument using HiSeq SBS Kit v4-H (catalogue #FC-401-4002) to perform paired-end 125-bp sequencing. The coverage was set as 90 Gb in HPDE cells expressing Mock and 180 Gb in cells expressing (P)RR. The software HiSeq Control Software v2.2.58 and Real Time Analysis v1.18.64 provided us with information on base calling and a quality score assigned to each base call.

Revised, pages 34-35, Methods, Whole -genome sequencing and mutation calling:

In accordance with the cBot User Guide Rev. L (Part #15006165), whole-genome libraries were prepared for cluster generation by cBot. To analyse sequences on the next-generation sequencer, the flow cells were clustered on the cBot using HiSeq PE Cluster Kit v4 cBot (Illumina; catalogue #PE-401-4001). DNA libraries were analysed on the Illumina HiSeq 2500 instrument using HiSeq SBS Kit v4-H (catalogue #FC-401-4002) to perform paired-end 125-bp sequencing. In HPDE-1/E6E7 cells expressing transient Mock and (P)RR and HPDE-1/E6E7 cells without transfection, libraries were prepared using the Tru Seq™ DNA PCR-Free Library Preparation Kit (Illumina; catalogue #FC-121-3001). Cluster generation and sequencing were prepared on a Novaseq 6000 system using Novaseq 6000 S2 Reagents (Illumina; catalogue #200012860). Pair-end 150-bp sequencing was performed. The coverage was set to 30 x for all the samples.

Point mutations, indels and SVs were detected as previously described¹⁶. We used HPDE-1/E6E7 cells without transfection as the reference and mutations were detected by comparing each Mock- and (P)RR-overexpressing HPDE-1/E6E7cell against the reference. Since mutations should have accumulated in HPDE-1/ E6E7 cells before transfection, we removed common mutations detected in 2 or more samples.

Original, pages 34-36, Methods, Sequence alignment and variant call:

The reference genome was based on the Genome Reference Consortium (ftp://ftp.ensembl.org/pub/release83/fasta/homo_sapiens/dna/Homo_sapiens.GRCh38.dna.primary_assembly.fa.gz) GRCh38.p5 supplemented with pEB Multi-Neo Target tag-C inserted with the open reading frame of ATP6ap2. Fastq files have been deposited with the National BioScience Database Center (NBDC) as file accession No. JGAS0000000143. After trimming the low-quality sequence reads, in which less than 50% of nucleotides had a quality value of 30, using the FASTX toolkit (http://hannonlab.cshl.edu/fastx_toolkit/) the fastq reads that passed these steps were mapped to GRCh38.p5 supplemented with the vector sequences using BWA-MEM in BWA ver. 0.7.12.10 with the default options; we removed PCR duplicates of the reads using the Mark Duplicates tool in Picard (<http://broadinstitute.github.io/picard/>). The mapped data were stored as BAM files.

For the alignment and variant calls, we attempted to compare the data analysed by two different software packages: GATK (The Genome Analysis Toolkit v2.5-2)⁶⁴ and Bcftools¹¹. We followed each recommended option in the processes for realignment, base quality assessment and recalibration of local insertion/deletion/single-nucleotide variation in GATK and used the default options for Bcftools. In both cases, we omitted the regions for which the coverage depth was less than 10. During the examination of variant call data, the variant calling process by the GATK toolkit under multiple CPUs caused fluctuation in SNP calling, with low quality, so we analysed all of the data using variant calls obtained by Bcftools. The accuracy of SNPs identified by Bcftools was also determined using the Genome-Wide Human SNP Array, which covered more than 906,600 SNPs (Thermo Fisher Scientific) under Perl scripts. We also utilized an integrative genome viewer to determine the quality of read alignments at those variant calls⁶⁵. Variant calls having a phred quality score of more than 10 were certified as data and annotated with snpEff⁶⁶ for gene ID, gene symbols, IDs of dbSNP 144 build and amino acid changes for nonsynonymous variants.

Revised:

Deleted.

Original, page 36, Methods, Identification of structural variation:

Structural variants (SVs) in HPDE cells expressing either Mock or (P)RR were called using the BreakDancer ver.1.1 software¹⁰ under the default options in the BAM files. To display the distribution of SVs in the whole genome, chromosomes were split into windows of specific sizes of 10, 100, 1,000, 10,000 and 100,000 bp. The SVs detected in each window were visualized as a Circos plot⁶⁷. In terms of DNA copy number variation (CNV), we determined the regions having CNVs in HPDE cells expressed by (P)RR, using HPDE expressed by Mock as a reference genome under CNV-seq⁶⁸.

Revised:

Deleted.

Change #24

As we have decided to delete the data for SNP genotyping in the exome array, we have also removed related methods.

Original, pages 36-37, Methods, SNP genotyping in the exome array:

SNP genotyping to investigate the presence of genetic diversification between passages in HPDE and HEK293 cells expressed by (P)RR was performed under an Infinium Exome-24 v1.2 BeadChip (Illumina; catalogue #WG-353-1204). This system has coverage of over 240,000 exonic variants without biologically significant associations. The call rate was more than 99.5% in all of the samples. The GenCall Score indicating the reliability of each genotype call met the standard. Information on the SNP probe gene annotation was obtained from

ftp://ussd-ftp.illumina.com/downloads/ProductFiles/HumanExome-12/HumanExome-12v1-2_A.annotated.txt.

Revised:

Deleted.

Change #25

We have added descriptions for beeswarm used for DNA fibre and comet assays and statistical analysis for direct cell counting.

Original, page 40, Methods, Statistical analysis:

Results are expressed as mean \pm SEM. We used one-way ANOVA with Scheffe's post hoc test to analyse the replication fork rates in the DNA fibre assay, DNA tails in the comet assay and telomere lengths in Flow-FISH. Cutoff was regarded as $P < 0.0001$ in both DNA fibre assay and comet assay. $P < 0.05$ was adopted in Flow-FISH.

Revised, page 38-39, Methods, Statistical analysis:

Results are expressed as mean \pm SEM. Beeswarm was depicted by R (The R Foundation for Statistical Computing Platform). We also used one-way ANOVA with Scheffe's post hoc test to analyse the replication fork rates in the DNA fibre assay and DNA tails in the comet assay and with Tukey-Kramer's post hoc test to analyse a cell proliferative ability in the deletion of each domain of (P)RR. Cutoff was regarded as $P < 0.0001$ in cell and nuclear area, DNA fibre assay and comet assay. $P < 0.05$ was adopted in direct cell counting and Flow-FISH

Change #26

We have made new section for “Data availability” to indicate the registration for the data of WGS and LC-MS/MS on the public database.

Revised, page 39, Methods, Data availability:

For WGS data, Fastq files were deposited with the National BioScience Database Center (NBDC) as file accession No. JGAS00000000143. For MS data, RAW data, peak lists and result files have been deposited as accession No. PXD 010107 with the ProteomeXchange Consortium⁶³ via the jPOST⁶⁴ partner repository as JPST000440.

Change #27

We have added four co-authors, i.e., Toru Furukawa denoted as T.O., Tsutomu Nakagawa denoted as T.N., Hiroyuki Ohsaki denoted as H.O. and Jing Hao Wong denoted as J.W. In the revised manuscript, S.Y. also designed the study. A.F. also designed the study and wrote the paper. K.T. also performed the analyses and interpretation of data.

Original, pages 40-41, Author Contributions:

Y.S., H.Y., D.Y., A.R., T.F., Y.F. and S.T. performed the experiments. Y.S., J.Y. and A.N. designed this study. Y.S., H.Y., J.Y., S.Y. and A.N. wrote the paper. Y.S., H.Y., J.Y., A.F., S.Y. and A.N. performed the analyses and interpretation of data. J.Y., K.T., Y.Y., H.Ko., M.T., S.Y. and H.Ki. contributed materials/analytical tools.

Revised, pages 39-40, Author Contributions:

Y.S., H.Y., D.Y., A.R., Ta.F., Y.F., S.T. and H.O. performed the experiments. Y.S., J.Y., S.Y., A.F. and A.N. designed this study. Y.S., H.Y., J.Y., J.W., S.Y., A.F. and A.N. wrote the paper. Y.S., K.T., H.Y., J.Y., A.F., H.O. and A.N. performed the analyses and interpretation of data. K.T., J.Y., H.Ko., T.M., Y.Y., H.Ki., S.Y., To.F. and T.N. contributed to materials/analytical tools.

Change #28

Associated with the change in the protocol of WGS analysis, we have deleted References #10, 11, 64, 65, 66, 67 and 68 of original manuscript. Additionally, we have removed Reference #20, 33, 34, 35, 36, 51, 52 and 53, followed by deleting a part of Discussion. For additional description in the Introduction, we have added new References #8, 9, 10, 12 and 13. References #18, 19, 20, 21 and 22 have also been added to the revised manuscript for the change in the results for WGS analysis. For the change of a part of Discussion, we have added Reference #41.

For the addition of HPDE-6/E6E7 cells, we have added Reference #16.

Change #29

We have completely changed the legend of Fig. 1.

Original, pages 46-47, Figure 1. Massive chromosomal rearrangements in HPDE cell population expressing (P)RR:

a, Cells transformed using an episomal vector inserted with ATP6ap2 encoding (pro)renin receptor [(P)RR]. b, Representative image of human pancreatic ductal epithelial (HPDE) cells expressing either Mock or (P)RR under a phase-contrast microscope. The morphology of cells expressing (P)RR becomes more diverse. Scale bar = 50 μ m. c, Proliferative ability of HPDE cells expressing either Mock or (P)RR. Proliferative ability became approximately 1.8-fold greater in cells expressing (P)RR than in Mock after 28 days of incubation (mean \pm SEM, N = 3 for each). d, Specific chromosomal rearrangements are shown at each window size in HPDE cells expressing either Mock or (P)RR. CTX: Intrachromosomal translocation; ITX: Interchromosomal translocation; INV: Inversion; INS: Insertion; DEL: Deletion. e, Venn diagrams comparing the total numbers of chromosomal rearrangements in Mock- and (P)RR-expressing cells. The total number of chromosomal rearrangements in the germline is indicated at the centre of the Venn diagrams. f, Upper: Circos plots showing the frequency and distribution of chromosomal rearrangements analysed by BreakDancer in cells expressing either Mock or (P)RR. Lower: Stack diagram of CTX, ITX, INV, INS and DEL in each chromosome of cells expressing either Mock or (P)RR. g, Number of SVs per 100 Mbp at a window size of 1,000 bp.

Revised, pages 45-46, Figure 1. (P)RR-expressing HPDE-1/E6E7 and HPDE-6/E6E7 cell population exhibited cellular atypia:

a, Upper: Vector constructs for a stable ATP6ap2 encoding (pro)renin receptor [(P)RR]. EBNA1 gene enables this vector to transfer to daughter cells in every cell division. Lower: Detection of (P)RR fused TARGET tag in human pancreatic ductal epithelial (HPDE)-1/E6E7 and HPDE-6/E6E7 cells at six- passage. b, Upper left: Representative image of HPDE-1/E6E7 cells expressing either Mock or (P)RR at six-passage under a phase-contrast microscope ($\times 50$). Upper right: The cell area in HPDE-1/E6E7 cells expressing either Mock or (P)RR. Averaged value of Mock cells is considered as 1 (N=100 for each, ***P<0.0001 vs. Mock). Lower left: Representative image of HPDE-6/E6E7 cells expressing either Mock or (P)RR at six-passage under a phase-contrast microscope ($\times 50$). Lower right: The cell area in HPDE-6/E6E7 cells expressing either Mock or (P)RR (N = 100 for each, ***P<0.0001 vs. Mock). c, Papanicolaou stain ($\times 400$). Upper left: Representative image of atypical nuclei in

HPDE-1/E6E7 cells expressing either Mock or (P)RR at six-passage. **Upper right:** The percentage of multinucleated cells and the nuclear area in HPDE-1/E6E7 cells expressing either Mock or (P)RR. Averaged value of Mock cells is considered as 1 (N = 100 for each, ***P<0.0001 vs. Mock). **Lower left:** Representative image of atypical nuclei in HPDE-6/E6E7 cells expressing either Mock or (P)RR at six-passage. **Lower right:** The percentage of multinucleated cells and the nuclear area in HPDE-6/E6E7 cells expressing either Mock or (P)RR. Averaged value of Mock cells is considered as 1 (N = 100 for each, ***P<0.0001 vs. Mock).

Change #30

We have completely changed the legend of Fig. 2.

Original, pages 47-48, Figure 2. Mutational landscape of HPDE cell population expressing (P)RR:

a, Venn diagrams comparing the total number of somatic mutations in Mock- with that in (P)RR-expressing cells. Total number of somatic mutations in the germline is indicated at the centre of the Venn diagrams. SNV: Single-nucleotide variation; INS: Insertion; DEL: Deletion. b, Upper left: Stack diagram of SNV, INS and DEL in each chromosome of cells expressing Mock. Lower left: Stack diagram of SNV, INS and DEL in each chromosome of cells expressing (P)RR. Upper right: The number of somatic mutations per 100 Kbp in Mock- and (P)RR-expressing cells. Lower right: The number of somatic mutations per 100 Kbp in (P)RR-expressing cells relative to that in Mock-expressing cells. c, Left: The number of somatic mutations of eight significant genes detected in PDAC patients is determined in cells expressing either Mock or (P)RR. Right: Stack diagram of eight significant genes in cells expressing either Mock or (P)RR. d, Left: The number of somatic mutations in BRCA pathway genes associated with the number of structural variations in pancreatic ductal adenocarcinoma (PDAC) patients in cells expressing either Mock or (P)RR. Right: Stack diagram of BRCA pathway genes in cells expressing either Mock or (P)RR. e, The number of mutated genes with annotations is shown in cells expressing either Mock or (P)RR.

Revised, pages 46-47, Figure 2. Genomic instability of HPDE-1/E6E7 cell population with transient and stable (P)RR expression:

a, **Left:** Vector constructs for stable and non-replicative transient ATP6ap2 encoding (pro)renin receptor [(P)RR] expression. Transfected cells were cultured with G418 for 21 days and analyzed after one passage. **Right:** Detection of (P)RR fused 10His tag in HPDE-1/E6E7 cells. b, **Upper:** Circos plot showing distribution of SVs in transient Mock -and (P)RR-expressing cell population. **Middle:** Number of each SV in transient Mock -and

(P)RR-expressing cell population. **Lower:** Total number of somatic mutations and mutated genes of the exome in transient Mock -and (P)RR-expressing cell population. **c, Upper:** Circos plot showing distribution of SVs in stable Mock -and (P)RR-expressing cell population. **Middle:** Number of each SV in stable Mock -and (P)RR-expressing cell population. **Lower:** Total number of somatic mutations and mutated genes of the exome in stable Mock -and (P)RR-expressing cell population.

Change #31

We have deleted original Fig.3.

Original, pages 48-49, Figure 3. Difference in clonal architecture between passages in cell population expressing (P)RR.

The number of single-nucleotide polymorphisms (SNPs) is determined by an exome array in 6- and 12-passage HPDE and HEK293 cells expressing (P)RR. The transfection of (P)RR fused TARGET tag is confirmed in 12-passage (a) HPDE and (b) HEK293 cells. Comparison of the number of mutated genes with annotations between 6- and 12-passage (a) HPDE and (b) HEK293 expressing (P)RR. The number of SNPs is increased in a passage-dependent manner. **c,** (P)RR-expressing cell population becomes more heterogeneous with the increase of passages, as shown by a schematic drawing.

Revised:

Deleted.

Change #32

We have moved the legend of Fig. 4 to Fig. 3.

Change #33

We have moved the legend of Fig. 5 to Fig. 4.

Change #34

We have moved the legend of Fig. 6 to Fig. 5 and added descriptions to Fig. 5 in the revised manuscript.

Original, pages 50-51, Figure 6. Dysfunction of genomic stability pathways by aberrant (P)RR expression in HPDE cells:

a, Ingenuity Pathway Analysis (IPA) for molecular functions of the downregulated molecules under (P)RR overexpression. b, Canonical pathways affected by (P)RR overexpression. c, Canonical pathways identified with high IPA statistical confidence. Underline indicates molecules confirmed by Western blot. d, Representative image of the downregulated molecules evaluated by Western blot. Consistent results are observed in three independent experiments. e, Constructs of deletion mutants in human (P)RR and confirmation of gene transfection in the vectors with each of Mock (M), FL(P)RR (FL), NTF of (P)RR (Δ C) and CTF of (P)RR (Δ N). Endogenous (P)RR was used as a loading control.

Revised, pages 48-49, Figure 5. Dysfunction of genomic stability pathways by aberrant (P)RR expression in HPDE-1/E6E7 cells:

a, Diseases expected from molecules downregulated by aberrant (P)RR expression. b, Ingenuity Pathway Analysis (IPA) for molecular functions of the downregulated molecules under (P)RR overexpression. c, Canonical pathways downregulated by (P)RR overexpression. d, Canonical pathways identified with high IPA statistical confidence. Underline indicates molecules confirmed by Western blot. e, Constructs of deletion mutants in human (P)RR and confirmation of gene transfection in the vectors with each of Mock (M), FL(P)RR (FL), NTF of (P)RR (Δ C) and CTF of (P)RR (Δ N). Endogenous (P)RR was used as a loading control. f, Expression of molecules involved in genomic stability pathways in cells with the deletion of each domain of (P)RR. Consistent results are obtained in three independent experiments. g, Activation of Wnt components. Consistent results are obtained in three independent experiments. h, Cell proliferative ability (mean \pm SEM, N = 3 for each, *P<0.05 vs. FL(P)RR, N.S., not significant).

Change #35

We have moved the legend of Fig. 7 to Fig. 6 and added descriptions to Fig. 6 in the revised manuscript.

Original, pages 51-52, Figure 7. Direct molecular binding of (P)RR with SWI/SNF-related, matrix-associated, actin-dependent regulator of chromatin, subfamily a, member 5 (SMARCA5):

e, Direct molecular binding of (P)RR with SMARCA5 under coimmunoprecipitation. MWM: molecular weight markers; Lys: lysates; rIgG: rabbit IgG. Consistent results are obtained in three independent experiments.

Revised, page 50, Figure 6. Direct molecular binding of (P)RR with SWI/SNF-related, matrix-associated, actin-dependent regulator of chromatin, subfamily a, member 5 (SMARCA5):

e, Binding of CTF of (P)RR with SMARCA5 under coimmunoprecipitation using insoluble nucleus. MWM: Molecular Weight Marker; Lys: Lysates; rIgG: rabbit IgG; IP: Immunoprecipitation; IB: Immunoblot. f, Expression of molecules responsible for genomic stability in (P)RR-expressing HPDE-1/E6E7 cells transfected with SMARCA5 siRNA. Consistent results are obtained in three independent experiments for Western blot and coimmunoprecipitation.

Change #36

We have moved the legend of Fig. 8 to Fig. 7.

Change #37

We have completely changed Fig. 1 and Fig. 2, and deleted Fig.3 in the revised manuscript. Please see the detailed information described in Changes # 7, 8, 9, 29, 30 and 31. We have updated several data shown in Fig. 5 and Fig. 6 of the revised manuscript. Please see the detailed information described in Changes # 10, 11, 34 and 35.

Change #38

We have removed Supplementary Table 1 to Table 6, because of the change in protocols for WGS and deletion of SNP array. Alternatively, we have added Supplementary Tables 1 -3 (data for WGS reanalyses) to the revised manuscript. Please see the detailed information described in Changes # 8 and 9. Table for the information obtained by LC-MS/MS has been moved to Supplementary Table 4. In Supplementary Table 5, we have updated the information of primers used for the detection of (P)RR tagged with 10His.

Change #39

We added several pictures showing atypical nuclei in (P)RR-overexpressing HPDE-1/E6E7 and HPDE-6/E6E7 cells, as shown in Supplementary Fig. 1. We have updated the data regarding WGS re-analysis in Supplementary Fig. 2. Please see the detailed information described in Change #8. In Supplementary Fig. 3 of the revised manuscript, we have indicated the data for generation of diverse atypical nuclei in (P)RR-expressing cell population. In Supplementary Fig. 4c of the revised manuscript, we have added the data for the related diseases expected from the upregulated molecules by (P)RR overexpression. Please see the detailed information in Change

#10. In Supplementary Fig. 5 of the revised manuscript, we have updated the data for the expression of the components involved in genomic stability pathways in HEK 293 cells with the deletion of (P)RR and in HPDE-1/E6E7 cells with seven-passage FL(P)RR, 10- and 20- passage Mock, respectively. Please see the detailed information in Change #10. In Supplementary Fig. 8 of the revised manuscript, we added the gating strategy for Flow-FISH in (P)RR-expressing HPDE-1/E6E7 cell population. In Supplementary Fig. 9 of the revised manuscript, we have changed from merged molecular network to three different separated molecular networks induced by (P)RR overexpression. In Supplementary Fig. 10d of the revised manuscript, we have updated the data for expression of the components involved in genomic stability pathways in (P)RR-overexpressing HEK293 cells transfected with SMARCA5 siRNA. Please see the detailed information in Change #11.

This is the end of the list of changes.

Thank you very much.

Reviewers' comments:

Reviewer #1 (Remarks to the Author):

I am generally greatly impressed by all the efforts made to improve the manuscript, but I regret to say that I remain unhappy with the response to no 6.

The authors say that they have preliminary data on knockdown to confirm specificity but that they cannot reveal it due to a pending patent application.

What I have always done in such a situation is to make a preliminary filing of the application for patent, which still allows new data to be added within 12 months, and then publish the paper (your data is patent protected by the initial filing). I think it is an important paper but it needs this evidence of specificity before it should be published, in my opinion - the claims made in the paper are big, and the bar for acceptance therefore needs to be set high

Reviewer #2 (Remarks to the Author):

Shibayama et. al. have extensively revised their study including conducting several additional experiments. I thank the authors for their thoughtful explanations and providing a comprehensive summary of their revision. I have the following comments/requests for clarification:

1. For experiments in figure 1 and supplementary figure 1, how many independent replicates for the transfections were used? Since WGS data seems to be an n=1 from each condition and the number of mice replicates are low, it would be important that this experiment with cell morphology readouts has been extensively repeated. Results from mock control should also be included in supplementary figure 1.
2. From the methods, it appears that co-immunoprecipitation experiments were conducted in HPDE-1/E6E7 cells and HEK293 cells. Which cell line is depicted in figure 6? Is there a clearer image for loading control available? Can figure label be changed to the actual loading control used?
3. From the methods it appears that two different siRNA constructs were used to for reducing SMARCA5 expression. Can results from the second construct also be included?
4. There seems to be a discrepancy in the text citations for the figures pertaining to the network analysis (page 15 -supplementary figure 9).
5. What is the source/supplier for the HPDE1 and HPDE6 cells?

Response to comments by reviewers

We thank the reviewers for their thoughtful review and helpful comments, which have guided the revision of the enclosed manuscript. In response to the various comments, we have made nine changes in the revised manuscript. We believe that the manuscript has been much improved.

To the comments by reviewer #1

Specific comment #1:

I am generally greatly impressed by all the efforts made to improve the manuscript, but I regret to say that I remain unhappy with the response to no 6. The authors say that they have preliminary data on knockdown to confirm specificity but that they cannot reveal it due to a pending patent application. What I have always done in such a situation is to make a preliminary filing of the application for patent, which still allows new data to be added within 12 months, and then publish the paper (your data is patent protected by the initial filing). I think it is an important paper but it needs this evidence of specificity before it should be published, in my opinion - the claims made in the paper are big, and the bar for acceptance therefore needs to be set high

Reply

Many thanks for this excellent suggestion. As we mentioned previously in reply to your original comment #6, we have already performed preliminary experiments to investigate whether the (P)RR siRNA rescues genomic stability. Array-CGH was performed to examine the effect of the (P)RR siRNA and a scrambled siRNA on DNA copy number variation in human PDAC PK-1 cells, and the data showed that the (P)RR siRNA significantly attenuates genome abnormality at all chromosome levels in these cells.

Because we intend to include these data in our patent application, we did not wish to include them in the manuscript. However, you kindly suggested that we file a preliminary application for a patent now, which would allow us to add more detailed data later (within 12 months of filing the preliminary application) and thus show these preliminary data in both the patent application and the manuscript. We agree that this is an ideal solution, and therefore contacted the Intellectual Property Department of Kagawa University to request that they expedite processing of the patent application. Unfortunately, however, they indicated that they are not able to respond to our request at this time, because the department's activities have been

suspended due to the effects of COVID-19. They also indicated that the patent firm that the university contracts with has suspended its activities as well. Indeed, all activities involving patent attorneys and lawyers in Japan have been suspended at this time, and only important cases are being prioritized.

Thus, although we did our best to enact the reviewer's suggestion, we are unfortunately not able to show the preliminary data in the revised manuscript given the current situation. However, we have included additional data demonstrating the effects of (P)RR expression on genomic stability in Fig. 2. We believe that these data support the concept that aberrant (P)RR expression induces genomic instability.

To the comments by reviewer #2

Specific comment #1:

For experiments in figure 1 and supplementary figure 1, how many independent replicates for the transfections were used? Since WGS data seems to be an n=1 from each condition and the number of mice replicates are low, it would be important that this experiment with cell morphology readouts has been extensively repeated. Results from mock control should also be included in supplementary figure 1.

Reply

First of all, we apologize for the lack of clarity in our initial description of this experiment. The vector, as shown in Fig. 1, contains the EBNA gene, which enables it to be transferred to the daughter cell at each round of cell division. Using this vector, we are therefore able to maintain a population of cells with a stable (P)RR expression level at every passage, as long as the transfected cells are grown in the presence of G418. As pointed out by the reviewer, only one transfection was performed.

We added this information to the legend for Fig. 1a (Change #5) as follows: "Construction of stably maintained vectors containing ATP6ap2, which encodes (pro)renin receptor [(P)RR]. The inclusion of the EBNA1 gene enables this vector to be transferred to daughter cells at each round of cell division."

Regarding the in vivo experiments, we used six kidneys from four animals. Two of the kidneys were used as the Mock injection group, and four of the kidneys were injected with (P)RR-expressing cells. The data consistently showed that the (P)RR-expressing HPDE-1/E6E7

cell began to form tumors, while visible expansion was entirely absent in the kidneys injected with Mock cells (Fig. 3b, c). In addition, all of the tissues formed by the injected (P)RR-expressing cells were composed of atypical cells with different shapes, atypical and swollen nuclei, distinctive nuclear bodies, and chromosomal abnormalities (Fig. 3d, e and Supplementary Fig. 3).

Although these changes were consistently observed in all four kidneys, the number of animals used was small, as noted by the reviewer. Therefore, we requested permission to repeat these experiments using the same protocol. However, our request was denied by the Animal Institute of Kagawa University, citing the need to minimize animal experiments given the COVID-19 situation in Japan. Indeed, since March 26 the animal facility has merely continued to house the animals that it was already caring for, and no new experimental protocols have been approved since that date.

Because additional animal experiments would have been extremely difficult to conduct under the current conditions, we decided to perform additional in vitro experiments. As shown in Fig. 1 and Supplementary Fig. 1, further analysis with Papanicolaou stain demonstrated consistent abnormalities such as multinucleation, enlarged and variable nuclear and cell sizes, and the presence of abnormal chromosomes in HPDE-1/E6E7 and HPDE-6/E6E7 cells. As suggested by the reviewer, we have also added images of the HPDE-1/E6E7 and HPDE-6/E6E7 control (Mock) cells to Supplementary Fig. 1. Thus, although we were not able to repeat the experiment with more animals given the current situation, we believe that these in vitro data combined with the previous in vivo data support the concept that aberrant (P)RR expression induces genomic instability in HPDE cells.

We have added images of the HPDE-1/E6E7 and HPDE-6/E6E7 control (Mock) cells to Supplementary Fig. 1 in the revised manuscript (Change #9).

Specific comment #2:

From the methods, it appears that co-immunoprecipitation experiments were conducted in HPDE-1/E6E7 cells and HEK293 cells. Which cell line is depicted in figure 6? Is there a clearer image for loading control available? Can figure label be changed to the actual loading control used?

Reply

Again, we apologize for the lack of clarity in our original description of this experiment. The co-immunoprecipitation data shown in Fig. 6e are from HPDE-1/E6E7 cells. In accordance with the reviewer's suggestion, we have inserted the names of each cell line used into Fig. 6e and Supplementary Fig. 10c. Furthermore, we removed the loading control images from Fig. 6e and Supplementary Fig. 10c and inserted images that show the actual loading in an unprocessed blot in the Supplementary information (the non-specific bands show the actual relative loading amounts).

We have inserted cell line labels into the co-immunoprecipitation images shown in Fig. 6e (Change #7) and Supplementary Fig. 10c (Change #9). To show the actual loading, we added an image of uncropped blot No.7 to the Supplementary information (Change #9).

Specific comment #3:

From the methods it appears that two different siRNA constructs were used to for reducing SMARCA5 expression. Can results from the second construct also be included?

Reply

We thank the reviewer for this comment. A combination of two siRNAs was used to enhance the efficiency of SMARCA5 knockdown. In the original version of this manuscript we showed data from experiments using a SMARCA5 siRNA #1 in (P)RR-overexpressing cells; however, we have since updated the manuscript to show the results from the same experiment using two different siRNA constructs.

We added the data showing the effect of SMARCA5 siRNA #2 alone on (P)RR-expressing cells to Fig. 6f (Changes #8), and have updated the Results, Methods, Figure legends (Changes #2, 4, and 6), and Supplementary Fig. 10d (Changes #2, 4, and 9) accordingly.

Specific comment #4:

There seems to be a discrepancy in the text citations for the figures pertaining to the network analysis (page 15 -supplementary figure 9).

Reply

We apologize for the confusion. All of the data regarding the molecular networks are shown in Fig. 6a, Supplementary Fig. 9, and Supplementary Table 4. We have corrected all of the text citations for these data in the revised manuscript.

In the revised manuscript, we added citations for all of the data pertaining to the molecular networks to the Results (Change #1).

Specific comment #5:

What is the source/supplier for the HPDE1 and HPDE6 cells?

Reply

Thank you for your valuable comment. Our HPDE-1 cells were transfected with the E6/E7 gene from human papilloma virus for the purpose of long-term culture and immortalization. The HPDE-1/E6E7 cell line was established by Dr. Furukawa (Am. J. Path., 1996), and the HPDE-6/E6E7 cell line was established by Dr. Ouyang (Am. J. Path., 2000).

In the revised manuscript, we have added the citation for each cell line to the 'Cell lines' subsection of the Methods (Change #3).

List of changes

Change #1

We cited all of the results regarding the molecular networks in the text.

1st Revision, page 15, Results, Direct molecular binding of (P)RR with SMARCA5 to form an ISWI chromatin remodelling complex

To investigate the genome integrity network responsible for transcriptional regulation, DNA replication, DNA repair and telomere maintenance, we performed a network analysis of IPA with filtered significant peptides under conditions of aberrant (P)RR expression. These analyses resulted in the identification of four different functional networks with significant Ingenuity scores, which were related to molecular functions such as gene expression, cellular assembly and organization, and DNA replication, recombination and repair (Supplementary Fig. 9 and Supplementary Table 4). These molecular networks could also be merged into a single molecular network, as they had several molecules in common (Supplementary Fig. 9 and Supplementary Table 4).

2nd Revision, page 14, Results, Direct molecular binding of (P)RR with SMARCA5

To investigate the genome integrity network responsible for transcriptional regulation, DNA replication, DNA repair and telomere maintenance, we performed a network analysis of IPA with filtered significant peptides under conditions of aberrant (P)RR expression. These analyses resulted in the identification of four different functional networks with significant Ingenuity scores, which were related to molecular functions such as gene expression, cellular assembly and organization, and DNA replication, recombination and repair (Fig. 6a, Supplementary Fig. 9 and Supplementary Table 4). These molecular networks could also be merged into a single molecular network, because they had several molecules in common (Fig. 6a, Supplementary Fig. 9 and Supplementary Table 4).

Change #2

We added a description of the two different SMARCA5 siRNAs that were used to the Results.

1st Revision, page 16, Results, Direct molecular binding of (P)RR with SMARCA5 to form an ISWI chromatin remodelling complex

Treatment with a SMARCA5 siRNA rescued the expression of molecules responsible for genomic stability pathways in (P)RR-expressing cells (Fig. 6f and Supplementary Fig. 10d)

2nd Revision, page 15, Results, Direct molecular binding of (P)RR with SMARCA5

Treatment with two different SMARCA5 siRNAs rescued the expression of molecules responsible for genomic stability pathways in (P)RR-expressing cells (Fig. 6f and Supplementary Fig. 10d).

Change #3

We added citations for the HPDE-1/E6E7 and HPDE-6/E6E7 cell lines to the 'Cell lines' subsection of the Methods.

1st Revision, page 23, Methods, Cell lines:

Immortalized HPDE-1/E6E7¹⁵ and HPDE-6/E6E7¹⁶ cells were cultured in Hu-Media KG2 (Kurabo, Osaka, Japan; catalogue #KK-2150S).

2nd Revision, page 22, Methods, Cell lines:

Immortalized HPDE-1/E6E7¹⁵ and HPDE-6/E6E7¹⁶ cells were cultured in Hu-Media KG2 (Kurabo, Osaka, Japan; catalogue #KK-2150S).

Change #4

We added the sequence of SMARCA5 siRNA#2 to the Methods.

1st Revision, page 25, Methods, Plasmid and siRNA transfection

In accordance with the manufacturer's instructions, FL(P)RR-, (P)RR-ΔN-, (P)RR-ΔC- or SMARCA5-expressing vector or an empty vector was transfected into HPDE-1/E6E7, HPDE-6/E6E7 and HEK293 cells using Lipofectamine 3000 (Thermo Fisher Scientific; catalogue #L3000008). Using Lipofectamine RNAiMAX (Thermo Fisher Scientific; catalogue #13778075), SMARCA5 Stealth RNAi™ siRNA was transfected into (P)RR-expressing cells in accordance with the recommended protocol. The primers used to generate the SMARCA5 Stealth RNAi™ siRNA were as follows: 5'-GGA GGC UUG UGG AUC AGA AUC UGA A-3' and 5'-UUC AGA UUC UGA UCC ACA AGC CUC C-3'. Stealth RNAi™ siRNA Negative Control Med GC (Thermo Fisher Scientific; catalogue #12935-300) was used as the scrambled control siRNA.

2nd Revision, page 24, Methods, Plasmid and siRNA transfection

In accordance with the manufacturer's instructions, FL(P)RR-, (P)RR-ΔN-, (P)RR-ΔC- or SMARCA5-expressing vector or an empty vector was transfected into HPDE-1/E6E7, HPDE-6/E6E7 and HEK293 cells using Lipofectamine 3000 (Thermo Fisher Scientific; catalogue #L3000008). Using Lipofectamine RNAiMAX (Thermo Fisher Scientific; catalogue #13778075), SMARCA5 Stealth RNAi™ siRNA was transfected into (P)RR-expressing cells in accordance with the recommended protocol. The primers used to generate SMARCA5 Stealth RNAi™ siRNA #1 were as follows: 5'-GGA GGC UUG UGG AUC AGA AUC UGA A-3' and 5'-UUC AGA UUC UGA UCC ACA AGC CUC C-3'. The primers used to generate SMARCA5 Stealth RNAi™ siRNA #2 were as follows: 5'-CAG GGA AGC UCU UCG UGU UAG UGA A-3' and 5'-UUC ACU AAC ACG AAG AGC UUC CCU G-3'. Stealth RNAi™

siRNA Negative Control Med GC (Thermo Fisher Scientific; catalogue #12935-300) was used as the scrambled control siRNA.

Change #5

1st Revision, pages 45, Figure 1. (P)RR-expressing HPDE-1/E6E7 and HPDE-6/E6E7 cell population exhibited cellular atypia:

a, Upper: Stably maintained vector containing ATP6ap2, which encodes (pro)renin receptor [(P)RR]. The inclusion of the EBNA1 gene enables this vector to be transferred to daughter cells at each round of cell division. Lower: Detection of TARGET-tagged (P)RR in human pancreatic ductal epithelial (HPDE)-1/E6E7 and HPDE-6/E6E7 cells at six passages.

2nd Revision, pages 44, Figure 1. (P)RR-expressing HPDE-1/E6E7 and HPDE-6/E6E7 cell population exhibited cellular atypia:

a, Upper: Stably maintained vector containing ATP6ap2, which encodes (pro)renin receptor [(P)RR]. The inclusion of the EBNA1 gene enables this vector to be transferred to daughter cells at each round of cell division. Lower: Detection of TARGET-tagged (P)RR in human pancreatic ductal epithelial (HPDE)-1/E6E7 and HPDE-6/E6E7 cells at six passages.

Change #6

We added a description of the use of two different SMARCA5 siRNAs.

1st Revision, page 50, Figure 6. Direct molecular binding of (P)RR with SWI/SNF-related, matrix-associated, actin-dependent regulator of chromatin, subfamily a, member 5 (SMARCA5):

f, Expression of molecules responsible for genomic stability in (P)RR-expressing HPDE-1/E6E7 cells transfected with SMARCA5 siRNA

2nd Revision, page 49, Figure 6. Direct molecular binding of (P)RR with SWI/SNF-related, matrix-associated, actin-dependent regulator of chromatin, subfamily a, member 5 (SMARCA5):

f, Expression of molecules responsible for genomic stability in (P)RR-expressing HPDE-1/E6E7 cells transfected with two different SMARCA5 siRNAs.

Change #7

We added the label 'HPDE-1/E6E7' to Fig. 6e.

Change #8

We added data regarding the effect of SMARCA5 siRNA #2 on (P)RR-expressing HPDE-1/E6E7 cells to Fig. 6f.

Change #9

We added several images of HPDE-1/E6E7 and HPDE-6/E6E7 cells from the Mock group to Supplementary Fig. 1. We added the label 'HEK293' to Supplementary Fig. 10c. We added data regarding the expression of molecules involved in genomic stability in (P)RR-expressing HEK293 cells transfected with SMARCA5 siRNA #2 to Supplementary Fig. 10d. Uncropped blots are shown in the last part of the Supplementary information. The actual loading for the co-immunoprecipitation blot is shown as unprocessed blot No. 7.

This concludes the list of changes made to the manuscript.

Thank you

REVIEWERS' COMMENTS:

Reviewer #2 (Remarks to the Author):

My concerns have been adequately addressed by the authors.